# Structure of Myosin VI/Tom1 complex reveals a cargo recognition mode of Myosin VI for tethering

Shichen Hu[1], Yujiao Guo[1], Yingli Wang[1], Ying Li[1], Tao Fu[1], Zixuan Zhou[1], Yaru Wang[1], Jianping Liu[1] & Lifeng Pan [1]

Myosin VI plays crucial roles in diverse cellular processes. In autophagy, Myosin VI can facilitate the maturation of autophagosomes through interactions with Tom1 and the autophagy receptors, Optineurin, NDP52 and TAX1BP1. Here, we report the high-resolution crystal structure of the C-terminal cargo-binding domain (CBD) of Myosin VI in complex with Tom1, which elucidates the mechanistic basis underpinning the specific interaction between Myosin VI and Tom1, and uncovers that the C-terminal CBD of Myosin VI adopts a unique cargo recognition mode to interact with Tom1 for tethering. Furthermore, we show that Myosin VI can serve as a bridging adaptor to simultaneously interact with Tom1 and autophagy receptors through two distinct interfaces. In all, our findings provide mechanistic insights into the interactions of Myosin VI with Tom1 and relevant autophagy receptors, and are valuable for further understanding the functions of these proteins in autophagy and the cargo recognition modes of Myosin VI.

[1] State Key Laboratory of Bioorganic and Natural Products Chemistry, Center for Excellence in Molecular Synthesis, Shanghai Institute of Organic Chemistry, University of Chinese Academy of Sciences, Chinese Academy of Sciences, 200032 Shanghai, China. Correspondence and requests for materials should be addressed to L.P. (email: panlf@sioc.ac.cn)

Autophagy is an essential self-digestion process involving the lysosomal degradations of damaged or harmful cytoplasmic components, including bulk protein aggregates, damaged organelles, and invasive pathogens in eukaryotic cells[1–3]. Owing to its critical roles in maintenance of cellular homeostasis and confrontation with different forms of cellular stress, autophagy plays a vital role in numerous physiologic processes such as cellular remodeling, embryogenesis and immune response, and defects in autophagy are associated with a large number of human diseases, such as cancer, heart diseases, and neurodegenerative diseases[2–6]. As a major subtype of autophagic response, macroautophagy (hereafter referred to as autophagy) relies on characteristic double-membraned vesicles named autophagosomes, to sequester and deliver undesired materials to lysosomes for degradations[7,8]. After enclosing of autophagic substrates, autophagosomes may undergo maturation by docking and fusion with endosomes to generate amphisomes, which then fuse with lysosomes to form autolysosomes for the ultimate degradations[8–10]. The fusion of endosome with nascent autophagosome is believed to provide autophagosome with additional molecular machinery that is required for subsequent lysosome fusion[2]. So far, many proteins have been identified to involve in this process such as Myosin VI, the endosomal protein Tom1, and the ubiquitin-binding autophagy receptors, NDP52, TAX1BP1, and Optineurin[9,11–15]. However, many of the detailed molecular mechanism underlying the functions of these proteins in promoting the formation of amphisome as well as the maturation of autophagosome are still not well understood.

Myosin VI is the only currently known unconventional myosin motor that transports cargoes along the actin filaments from the plus end to the minus end[16–19]. As a unique myosin motor, Myosin VI can not only serve as a processive dimer or oligomer for vesicular transport and cargo sorting, but also function as a nonprocessive monomeric anchor or tether to link its binding cargoes with cytoskeletons for the establishment and maintenance of highly specialized cellular structures[16,18–23]. Therefore, Myosin VI plays essential roles in a range of cellular processes such as clathrin-mediated endocytosis, autophagy, exocytosis, and the development and maintenance of the stereocilia of hair cells[12,18,22,24–28]. Structurally, Myosin VI contains an N-terminal motor domain that can bind to and walk along the actin filaments in a reverse-direction[14], a neck region (also named as the lever arm) assembled by a unique reverse gear and an IQ motif, both of which can specifically bind to calmodulin[29], a 3-helix bundle region followed by a highly charged single α-helix region (SAH), and a C-terminal globular cargo-binding domain (CBD) that mainly includes two unique protein–protein interaction modules, the RRL motif and the extreme C-terminal cargo-binding module containing the WWY motif (hereafter referred to as the C-terminal CBD) (Fig. 1a). Several adaptor proteins including Disabled-2 (Dab2), LMTK2, Tom1, GIPC, NDP52, TAX1BP1, Optineurin as well as the ubiquitin proteins, have been implicated in the interactions with the C-terminal globular cargo-binding region of Myosin VI, thus endowing Myosin VI with specific subcellular localizations and functions[12,28,30–35]. For instance, Dab2, an endocytic adaptor protein, is able to bridge Myosin VI to clathrin-coated vesicles at the early stages of endocytosis[21,23,30]. Strikingly, previous biochemical and structural studies have well established that Myosin VI can undergo a cargo-binding induced dimerization or oligomerization to form a dimeric or oligomeric motor for processive walking along the actin filaments[21,23,36–38]. However, how Myosin VI serves as a nonprocessive monomeric motor to associate with cargoes for fulfilling an anchoring or tethering role is still largely unknown.

Tom1 together with its two paralogs, TomL1 and TomL2, belongs to a smaller subfamily, and was initially identified as a

binding target of Myb1 and later was proved as a part of the endosomal sorting complex (ESCRT-0 complex) required for transport[22,39,40]. Tom1 contains an N-terminal VHS domain, a central GAT domain that can specifically interact with ubiquitin and Tollip[41], a clathrin-binding motif and an ill-characterized C-terminal region (Fig. 1a). A previous study revealed that Tom1 can directly interact with Myosin VI through its uncharacterized C-terminal region, and was proved to be a crucial adaptor to mediate endosomal localization of Myosin VI[12]. Meanwhile, a proportion of the autophagy receptors, NDP52, TAX1BP1, and Optineurin, which are specifically located at the outer surface of autophagosome, can also associate with Myosin VI through their ubiquitin-binding regions, the C-terminal tandem zinc-fingers of NDP52 and TAX1BP1, as well as the UBAN region of Optineurin[12,15,42–44] (Fig. 1a). Importantly, Myosin VI in concert with Tom1 and the autophagy receptor NDP52, TAX1BP1, or Optineurin, can promote the fusion of autophagosomes with endosomes and facilitate the maturation of autophagosomes[12,15]. However, due to the lack of a detailed structural investigation, the precise binding modes of Myosin VI with Tom1 and these autophagy receptors remain elusive.

In this study, we systematically characterize the interactions of Myosin VI with Tom1 and three autophagy receptors, NDP52, TAX1BP1, and Optineurin, and discover that the C-terminal region of Tom1 contains a conserved Myosin VI-binding motif shared by all Tom1 family members, which can specifically interact with the C-terminal CBD of Myosin VI to form a stable heterodimeric complex. The determined high-resolution structure of Myosin VI/Tom1 complex uncovers that the C-terminal CBD of Myosin VI adopts a unique molecular mechanism to recognize Tom1, and in contrast to that of Dab2, the binding of Tom1 by the C-terminal CBD of Myosin VI is unable to induce the dimerization of Myosin VI. In addition, we demonstrate that Myosin VI may function as a tether to simultaneously interact with Tom1 and the autophagy receptor NDP52, TAX1BP1, or Optineurin, forming different ternary complexes for promoting autophagosome maturation. In all, our findings not only provide mechanistic insights into the interactions of Myosin VI with Tom1 and autophagy receptors, NDP52, TAX1BP1, and Optineurin, but also provide a paradigm for understanding the monomeric cargo recognition mode of Myosin VI for tethering.

## Results

**Mapping the interaction regions of Myosin VI and Tom1.** A previous study showed that the C-terminal uncharacterized region of Tom1 and the C-terminal globular cargo-binding domain of Myosin VI are responsible for the specific interaction between Tom1 and Myosin VI in cells[12]. In order to uncover the detailed molecular mechanism underpinning the interaction of Myosin VI and Tom1, we firstly sought to map out the precise binding regions of Myosin VI and Tom1. Based on the sequence conservation and secondary structure prediction, we chose four Tom1 fragments (residues 215–493, 215–392, 392–493, and 392–463) covering different C-terminal regions of Tom1 (Supplementary Fig. 1b), and then purified these proteins and conducted analytical gel filtration chromatography-based analyses to test their interactions with the Myosin VI globular cargo-binding domain (residues 1032–1285) (Supplementary Fig. 2a–d). Our results showed that except for Tom1(215–392), the Tom1(215–493), Tom1(392–493), and Tom1(392–463) fragments can directly bind to Myosin VI(1032–1285) (Supplementary Fig. 2a–d), indicating that the Tom1(392–463) fragment contains essential residues for the interaction with Myosin VI. Further biochemical analyses showed that the C-terminal CBD (residues 1157–1285) rather than the N-terminal

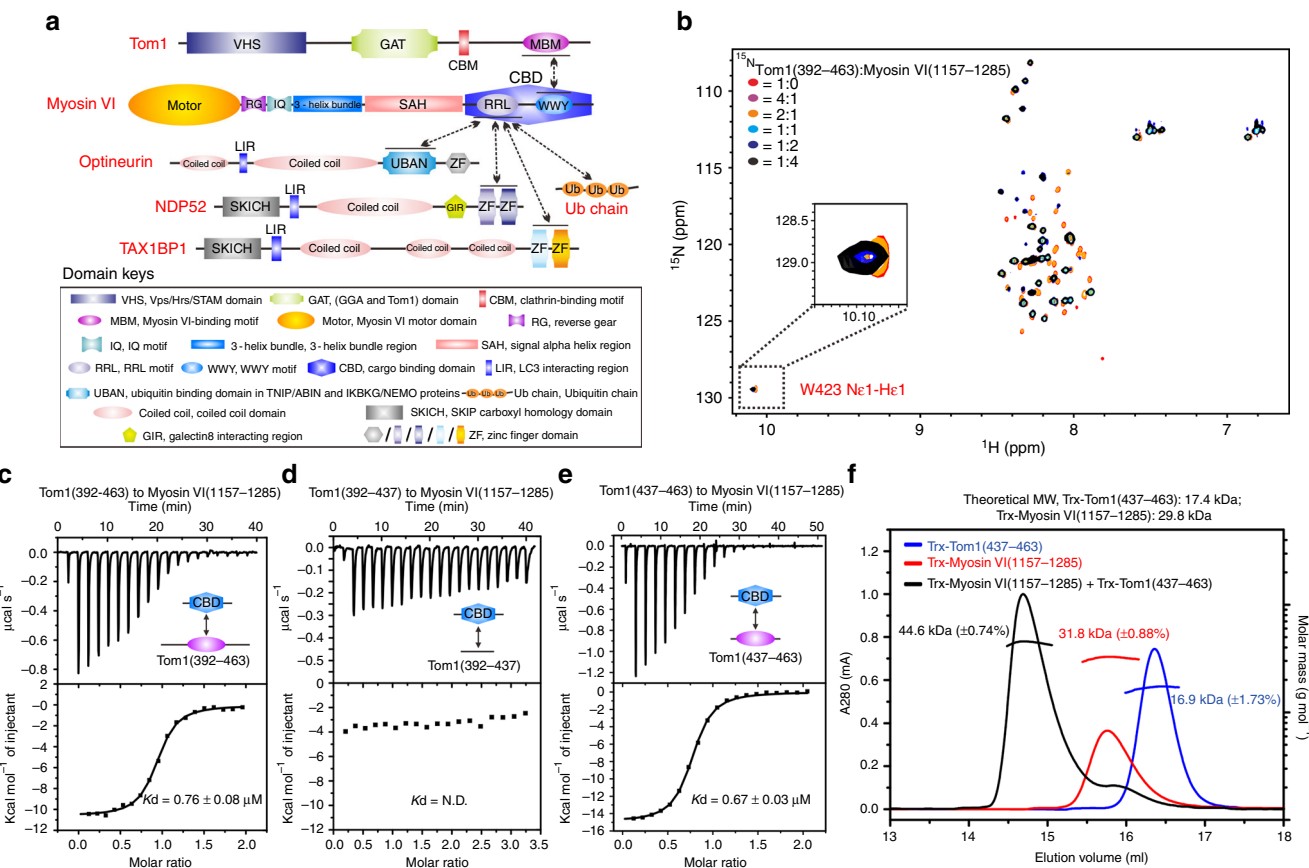

**Fig. 1** Biochemical analyses of the interaction between Myosin VI and Tom1. **a** A schematic diagram showing the domain arrangements of Myosin VI, NDP52, TAX1BP1, and Optineurin. In this drawing, domains involved in the protein–protein interaction are highlighted with black lines, and the relevant interactions between two proteins are indicated by two-way arrows. **b** Superposition plots of the $^1$H-$^{15}$N HSQC spectra of Tom1(392–463) titrated with the un-labeled C-terminal CBD of Myosin VI proteins at different molar ratios. For clarity, the insert shows the enlarged view of a unique peak corresponding to the side chain of Tom1 W423 residue in the overlaid $^1$H-$^{15}$N HSQC spectra. **c–e** ITC-based measurements of the binding affinities of the C-terminal CBD of Myosin VI with Tom1(392–463) (**c**), Tom1(392–437) (**d**), and Tom1(437–463) (**e**). Kd values are the fitted dissociation constants with standard errors, when using the one-site binding model to fit the ITC data. 'N.D.' stands for that the Kd value is not detectable. Source data are provided as a Source Data file. **f** Overlay plots of the multi-angle light-scattering data of the C-terminal CBD of Myosin VI, Tom1(437–463), and the C-terminal CBD of Myosin VI in complex with Tom1(437–463). The derived molecular masses of the C-terminal CBD of Myosin VI and Tom1(437–463) are shown in red and in blue, respectively, while the derived molecular mass of the C-terminal CBD of Myosin VI and Tom1(437–463) complex is shown in black. The molecular masses errors are the fitted errors obtained from the data analysis software, and are showed in the brackets. The results clearly demonstrate that the C-terminal CBD of Myosin VI and Tom1(437–463) both form a stable monomer and may interact with each other to form a 1:1 stoichiometric complex in solution. Source data are provided as a Source Data file

RRL motif region (residues 1032–1157) of the Myosin VI CBD is required for the interaction with Tom1(392–463) (Supplementary Fig. 2e, f). Next, we took advantage of the powerful nuclear magnetic resonance (NMR) spectroscopy to further characterize the interaction of Tom1(392–463) with the C-terminal CBD of Myosin VI. Titrations of $^{15}$N-labeled Tom1 (392–463) with un-labeled C-terminal CBD of Myosin VI proteins showed that many peaks in the $^1$H-$^{15}$N HSQC spectrum of Tom1(392–463) underwent significant peak-broadenings or chemical shift changes (Fig. 1b), confirming that Tom1 (392–463) can directly bind to the C-terminal CBD of Myosin VI. Notably, further detailed analysis revealed that the peak corresponding to the side chain of W423 residue in the $^1$H-$^{15}$N HSQC spectrum of Tom1(392–463), which contains only one tryptophan residue (Supplementary Fig. 1b), showed little changes when titrated with the C-terminal CBD of Myosin VI (Fig. 1b), suggesting that, in contrast to a previous report[45], the conserved $^{420}$IEXWL$^{424}$ motif of Tom1 is not directly involved in the Myosin VI-binding. To further narrow down the Myosin VI-binding region of Tom1, we carried out a quantitative

isothermal titration calorimetry (ITC)-based assay using the C-terminal CBD of Myosin VI and eight different Tom1 fragments, Tom1(215–493), Tom1(215–392), Tom1(392–493), Tom1(437–493), Tom1(464–493), Tom1(392–463), Tom1 (392–437), and Tom1(437–463) (Supplementary Fig. 3a–e and Fig. 1c–e). The ITC results revealed that the Tom1(215–493), Tom1(392–493), Tom1(437–493), Tom1(392–463) and Tom1 (437–463) fragments bind to the C-terminal CBD of Myosin VI with similar Kd values (Supplementary Fig. 3a, c, d and Fig. 1c, e and Table 1), while Tom1(215–392), Tom1(464–493) and Tom1 (392–437) cannot interact with the C-terminal CBD of Myosin VI (Supplementary Fig. 3b, e and Fig. 1d and Table 1). Therefore, the Myosin VI-binding motif of Tom1 is located within the Tom1(437–463) fragment (hereafter referred to as Tom1 MBM). Finally, using analytical gel filtration chromatography, multi-angle light-scattering and analytical ultracentrifugation analyses, we further elucidated that Tom1 MBM and the C-terminal CBD of Myosin VI both form monomers in solution, and importantly, they can interact with each other to form a 1:1 stoichiometric complex (Fig. 1f and Supplementary Fig. 3f, g).

**Table 1 Statistics of ITC results between different variants of Myosin VI and Tom1**

| Myosin VI | Tom1 | $Kd$ (µM) | $N$ | $\triangle H$ (kcal/mol) | $\triangle G$ (kcal/mol) | $-T\triangle S$ (kcal/mol) |
|---|---|---|---|---|---|---|
| 1157–1285 | 215–493 | 2.13 ± 0.12 | 1.44 ± 0.01 | −7.61 ± 0.09 | −7.73 | −0.12 |
| 1157–1285 | 215–392 | N.D. | N.D. | N.D. | N.D. | N.D. |
| 1157–1285 | 392–493 | 0.93 ± 0.09 | 1.12 ± 0.01 | −9.90 ± 0.11 | −8.23 | 1.68 |
| 1157–1285 | 392–463 | 0.76 ± 0.08 | 0.91 ± 0.01 | −10.60 ± 0.16 | −8.35 | 2.21 |
| 1157–1285 | 392–437 | N.D. | N.D. | N.D. | N.D. | N.D. |
| 1157–1285 | 437–493 | 0.76 ± 0.04 | 0.83 ± 0.01 | −12.70 ± 0.11 | −8.35 | 4.31 |
| 1157–1285 | 464–493 | N.D. | N.D. | N.D. | N.D. | N.D. |
| 1157–1285 | 437–463 | 0.67 ± 0.03 | 0.74 ± 0.01 | −14.90 ± 0.10 | −8.42 | 6.53 |
| 1157–1285 | 437–463(L448E) | N.D. | N.D. | N.D. | N.D. | N.D. |
| 1157–1285 | 437–463(R451E) | N.D. | N.D. | N.D. | N.D. | N.D. |
| 1157–1285(W1193A) | 437–463 | N.D. | N.D. | N.D. | N.D. | N.D. |
| 1157–1285(E1207R) | 437–463 | N.D. | N.D. | N.D. | N.D. | N.D. |

The $Kd$ values are the fitted dissociation constants with standard errors, when using the one-site binding model to fit the ITC data. $N$ stands for the binding stoichiometry, $\triangle H$ shows the enthalpy change, $\triangle S$ shows the entropy change. These values were obtained from ITC experiments. While $\triangle G$ shows the Gibbs energy change that is defined as: $\triangle G = \triangle H - T^{*}\triangle S$ ($T = 298$ K). 'N.D.' stands for that the $Kd$ value is not detectable

**Table 2 Data collection and refinement statistics**

| Data set | Myosin VI/Tom1 complex |
|---|---|
| Data collection | |
| Space group | $P12_{1}1$ |
| Unit cell parameters | |
| $a, b, c$ (Å) | 40.73, 74.88, 50.87 |
| $\alpha, \beta, \gamma$ (°) | 90, 106.34, 90 |
| Wavelength (Å) | 0.97774 |
| Resolution range (Å) | 50.00–1.80 (1.83–1.80) |
| Number of total reflections | 177,263 |
| Number of unique reflections | 26915 |
| Redundancy | 6.60 (6.20) |
| $I/\sigma I$ | 28.14 (4.74) |
| Completeness (%) | 98.50 (98.50) |
| $R_{merge}$ (%) [a] | 5.60 (35.40) |
| Structure refinement | |
| Resolution (Å) | 37.44–1.80(1.87–1.80) |
| $R_{work}$[b]/$R_{free}$[c] (%) | 17.66/23.35 |
| Number of reflections | |
| Working set | 25,194 |
| Test set | 1252 |
| $B$ factor (Å$^{2}$) | |
| Average | 31.37 |
| Protein | 30.99 |
| RMSD bonds (Å) | 0.01 |
| RMSD angles (°) | 1.16 |
| Number of non-hydrogen atoms | |
| Protein | 2281 |
| Ligand | 0 |
| Water | 249 |
| Ramachandran plot (%) | |
| Most favored | 99.24 |
| Additionally allowed | 0.76 |
| Outliers | 0 |

[a]$R_{merge} = \Sigma |I-<I>|/\Sigma<I>$, where $I$ is the intensity of the measured reflection,$<I>$ is the mean intensity of all symmetry-related reflections. Values in parentheses are for highest-resolution shell

[b]$R_{work} = \Sigma ||F_{obs}| - |F_{cal}||/\Sigma|F_{obs}|$, where $F_{obs}$ and $F_{cal}$ are observed and calculated structure factors

[c]$R_{free} = \Sigma_{T}||F_{obs}| - |F_{cal}||/\Sigma_{T}|F_{obs}|$, where $T$ is a randomly chosen test data set of about 5% of the total reflections and set aside prior to refinement. Values in parentheses are for the highest-resolution shell

**Overall structure of the Myosin VI/Tom1 complex**. To further elucidate the mechanistic basis underlying the specific interaction between the C-terminal CBD of Myosin VI and Tom1 MBM, we sought to determine their complex structure. We purified the C-terminal CBD of Myosin VI and Tom1 MBM complex, and

successfully obtained good crystals that diffracted to 1.80 Å resolution. Using the molecular replacement method with the modified structure of the C-terminal CBD of Myosin VI in the Myosin VI/Dab2 complex (PDB ID: 3H8D), we managed to solve the Myosin VI/Tom1 complex structure (Table 2). In the final complex structural model, an asymmetric unit contains two C-terminal CBD of Myosin VI/Tom1 MBM complexes (Supplementary Fig. 4a), each of which has a 1:1 stoichiometry and is composed of one C-terminal CBD of Myosin VI and one Tom1 MBM molecule (Supplementary Fig. 4a and Fig. 2a), consistent with our aforementioned biochemical analyses (Fig. 1f and Supplementary Fig. 3g). The overall structures of two complex molecules in the asymmetric unit are essentially the same, except that the flexible extreme N-terminal α-helix of Myosin VI in one complex molecule is unsolved due to the lack of electron density (Supplementary Fig. 4b). In the complex structure, the C-terminal CBD of Myosin VI features an architecture assembled by a 4-stranded antiparallel β-sheet packing with five α-helices (Fig. 2a), and particularly, one surface of the 4-stranded β-sheet is covered by three helices (α3–α5), while the other side is capped by the short α2 helix (Fig. 2a). Intriguingly, Tom1 MBM in the complex structure mainly forms a continuous α-helix (Supplementary Fig. 4a–c), and packs extensively with the solvent-exposed side of the 4-stranded β-sheet of Myosin VI containing the α2 helix (hereafter referred to as site I), burying a total of ~827 Å$^{2}$ surface area (Fig. 2b). Further structural comparison revealed that the C-terminal CBD of Myosin VI in the Myosin VI/Tom1 complex structure adopts a similar overall conformation to that of the apo-form protein, except for the extreme N-terminal and C-terminal α-helices, as well as the region linking β1 and β2 (Supplementary Fig. 4d).

**The molecular interface of the Myosin VI/Tom1 complex**. Further detailed structure analyses of the binding interface of Myosin VI/Tom1 complex revealed that the specific interaction between the C-terminal CBD of Myosin VI and Tom1 MBM is mainly mediated by extensive hydrophobic and polar interactions (Fig. 3a, b). In particular, the hydrophobic side chains of F444, F447, L448, and A452 of Tom1 occupy a hydrophobic groove formed by the side chains of I1173, F1175, W1193, C1227, L1229, T1234, and L1236 from Myosin VI, and concurrently, the hydrophobic side chains of Tom1 A454, A455, L458, P459, and L461 residues pack against a hydrophobic patch formed by the side chains of P1213 and I1215 together with the aliphatic side chain of K1212 from Myosin VI (Fig. 3a, c). In addition, the polar side chain groups of S440, R451, N460, S462, together with the

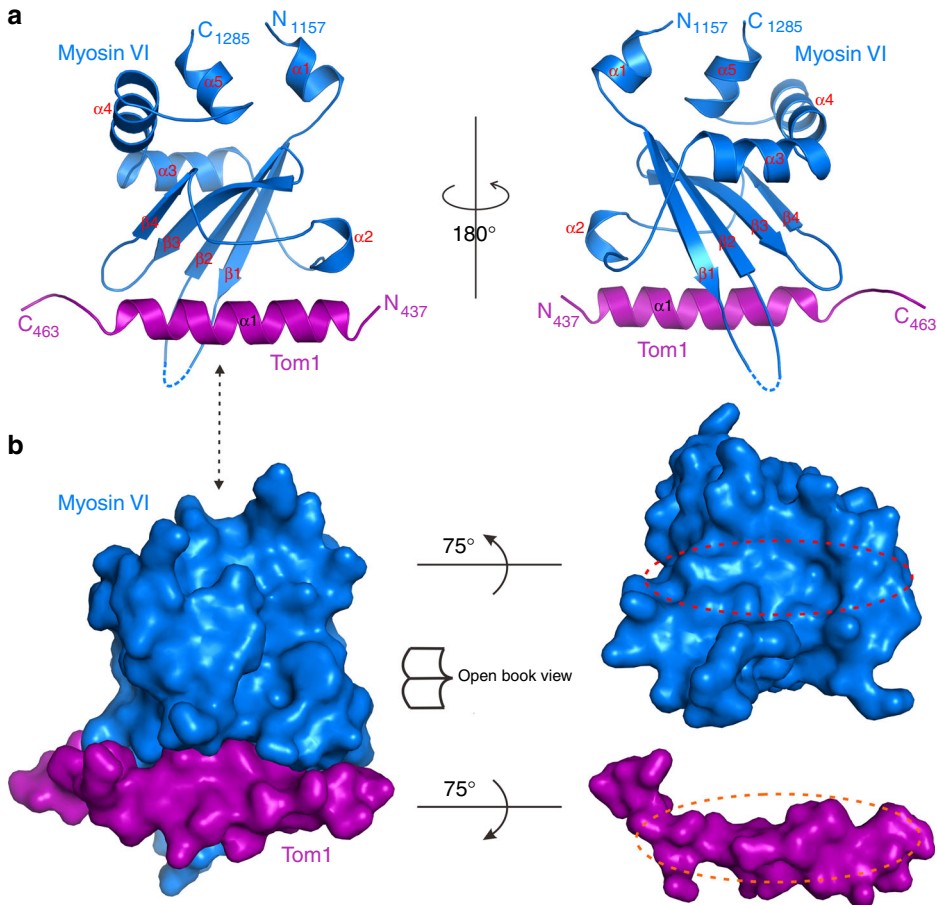

**Fig. 2** The overall structure of Myosin VI/Tom1 complex. **a** Ribbon diagram showing the overall structure of the C-terminal CBD of Myosin VI and Tom1 MBM complex. In this drawing, the C-terminal CBD of Myosin VI is shown in blue, and Tom1 MBM in magenta. **b** Surface representations showing the overall architecture of Myosin VI/Tom1 complex (left panel), and the open-book view of the binding interface between Myosin VI and Tom1 (right panel) with the same color scheme as in **a**

backbone carboxyl group of P459 and the backbone amine group of L461 from Tom1 interact with the D1211, K1212, E1225, and E1233 residues of Myosin VI to form six highly specific hydrogen bonds (Fig. 3c). Moreover, the Myosin VI/Tom1 complex is further stabilized by two Arg-Glu pair ($Arg1238_{Myosin\ VI}$-$Glu441_{Tom1}$ and $Arg451_{Tom1}$-$Glu1207_{Myosin\ VI}$) of salt bridges (Fig. 3c). In accordance with their important structural roles, all of these key residues of Tom1 and Myosin VI involved in the binding interface are highly conserved across different eukaryotic species (Supplementary Fig. 1). Using the ITC analysis, we further verified the specific interactions between Myosin VI and Tom1 observed in the complex structure. In line with our structural data, the ITC results showed that point mutations of key interface residues either from Myosin VI or Tom1, such as the W1193A, E1207R mutations of Myosin VI, or the L448E, R451E mutations of Tom1, all completely disrupted the specific interaction between the C-terminal CBD of Myosin VI and Tom1 MBM (Supplementary Fig. 5 and Table 1). Notably, further structure-based sequence alignment analysis showed that the key Myosin VI-binding interface residues of Tom1 can be also found in its two close homologs, TomL1 and TomL2 (Fig. 3d). Therefore, the Tom1 family proteins, including Tom1, TomL1, and TomL2, likely share a general binding mode to interact with Myosin VI.

**Myosin VI adopts different binding modes for Tom1 and Dab2.** In addition to Tom1, the C-terminal CBD of Myosin VI was also reported to interact with Dab2 and LMTK2[23,31,32]. In particular,

the interaction of Myosin VI and Dab2 was well structurally characterized in a previous elegant study[21], which elucidated that the C-terminal CBD of Myosin VI can simultaneously interact with two helical segments (αA and αB) of Dab2 by two distinct binding sites (site I and site II) located at the opposite sides of the C-terminal CBD of Myosin VI to undergo a Dab2-mediated dimerization (Fig. 4a). Strikingly, in contrast to Dab2, Tom1 can only occupy the site I of the C-terminal CBD of Myosin VI, and is unable to induce the dimerization of Myosin VI (Fig. 4a). Moreover, further detailed structural comparison revealed that although Myosin VI uses the site I to interact with both Tom1 and the αA segment of Dab2, it employs some different interface residues (Fig. 4a, b). Specifically, the I1173, F1175, W1193, C1227, L1229, E1233, T1234, and L1236 residues are involved in both Tom1 and Dab2 interactions, while the I1176, P1178, Q1205 residues and the E1207, D1211, K1212, P1213, I1215, E1225, R1238 residues are only specific for the Dab2-binding and Tom1-binding, respectively (Fig. 4b and Supplementary Fig. 1a). Conversely, the corresponding residues of Tom1 and Dab2 involved in the interactions with the C-terminal CBD of Myosin VI are also quite different, despite that the FXXF/Y motif is found in both Tom1 and Dab2 (Fig. 4b, c). Interestingly, careful sequence alignment analysis of the Myosin VI-binding regions in Tom1 and Dab2 showed that some residues in the αB segment of Dab2, which are critical for interaction with the site II of the C-terminal CBD of Myosin VI, can be also found in the extreme C-terminal region of Tom1 (Fig. 4c). However, further NMR-based analyses using $^{15}N$-labeled C-terminal CBD

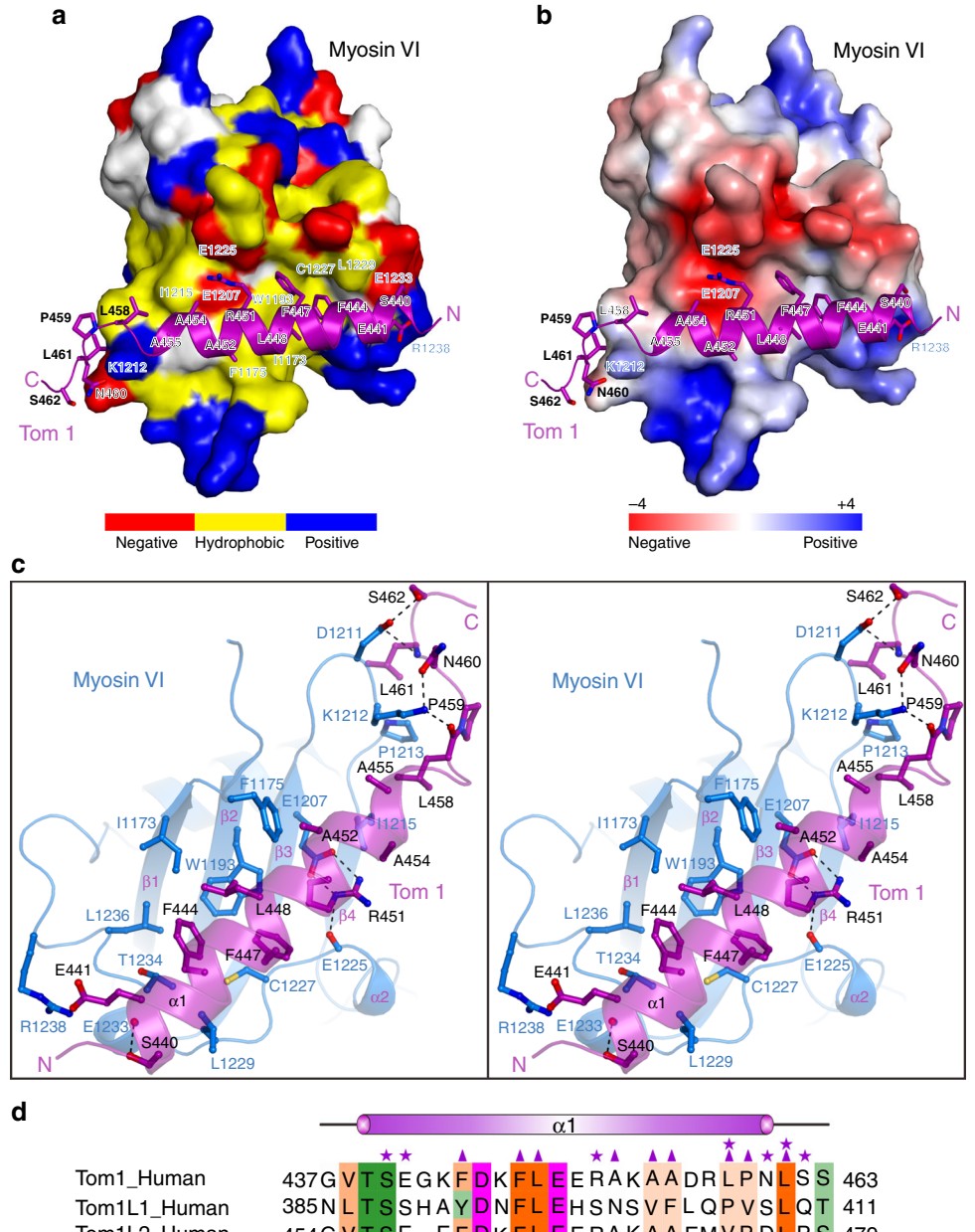

**Fig. 3** The molecular interface of Myosin VI and Tom1 complex. **a** The combined surface representation and the ribbon-stick model showing the hydrophobic binding interface between the C-terminal CBD of Myosin VI and Tom1 MBM. In this presentation, the C-terminal CBD of Myosin VI is shown in the surface model and Tom1 MBM in the ribbon-stick model. Particularly, in the surface model of the C-terminal CBD of Myosin VI, the hydrophobic amino acid residues are drawn in yellow, the positively charged residues in blue, the negatively charged residues in red, and the uncharged polar residues in gray. **b** The combined surface charge representation and the ribbon-stick model showing the charge-charge interactions between the C-terminal CBD of Myosin VI and Tom1 MBM. **c** Stereo view of the ribbon-stick model showing the detailed interactions between the C-terminal CBD of Myosin VI and Tom1 MBM. The hydrogen bonds and salt bridges involved in the binding are shown as dotted lines. **d** Structure-based sequence alignment of Tom1 MBM with the corresponding regions of TomL1 and TomL2. In this structure-based sequence alignment, the conserved hydrophobic residues, polar neutral residues, positively charged residues, and negatively charged residues are colored in orange, green, blue, and magenta, respectively. Interface residues of Tom1 that are involved in the polar interactions and hydrophobic interactions with the C-terminal CBD of Myosin VI in the Myosin VI/Tom1 complex are further labeled with magenta stars and magenta triangles, respectively

of Myosin VI titrated with un-labeled Tom1(392–463) or Tom1 (392–493) that includes the extreme C-terminal region of Tom1, showed that the two $^1$H-$^{15}$N HSQC spectra of the C-terminal CBD of Myosin VI saturated with excess amounts of Tom1 (392–463) and Tom1(392–493) are very similar (Supplementary Fig. 6a). Particularly, the NMR peak corresponding to the side chain of Myosin VI W1192 residue that is located at the site II of the C-terminal CBD of Myosin VI, displayed little chemical shift

changes in the presences of Tom1(392–463) and Tom1(392–493) (Supplementary Fig. 6b), but in contrast, it showed significant chemical shift changes when binding to Dab2[21]. In addition, our NMR titration experiment using $^{15}$N-labeled Tom1(392–493) titrated with the un-labeled C-terminal CBD of Myosin VI revealed that the MBM region of Tom1 (residues 440–462) shows significant dose-dependent peak-broadenings, while the region C-terminal to the MBM of Tom1 (residues 463–493) displays

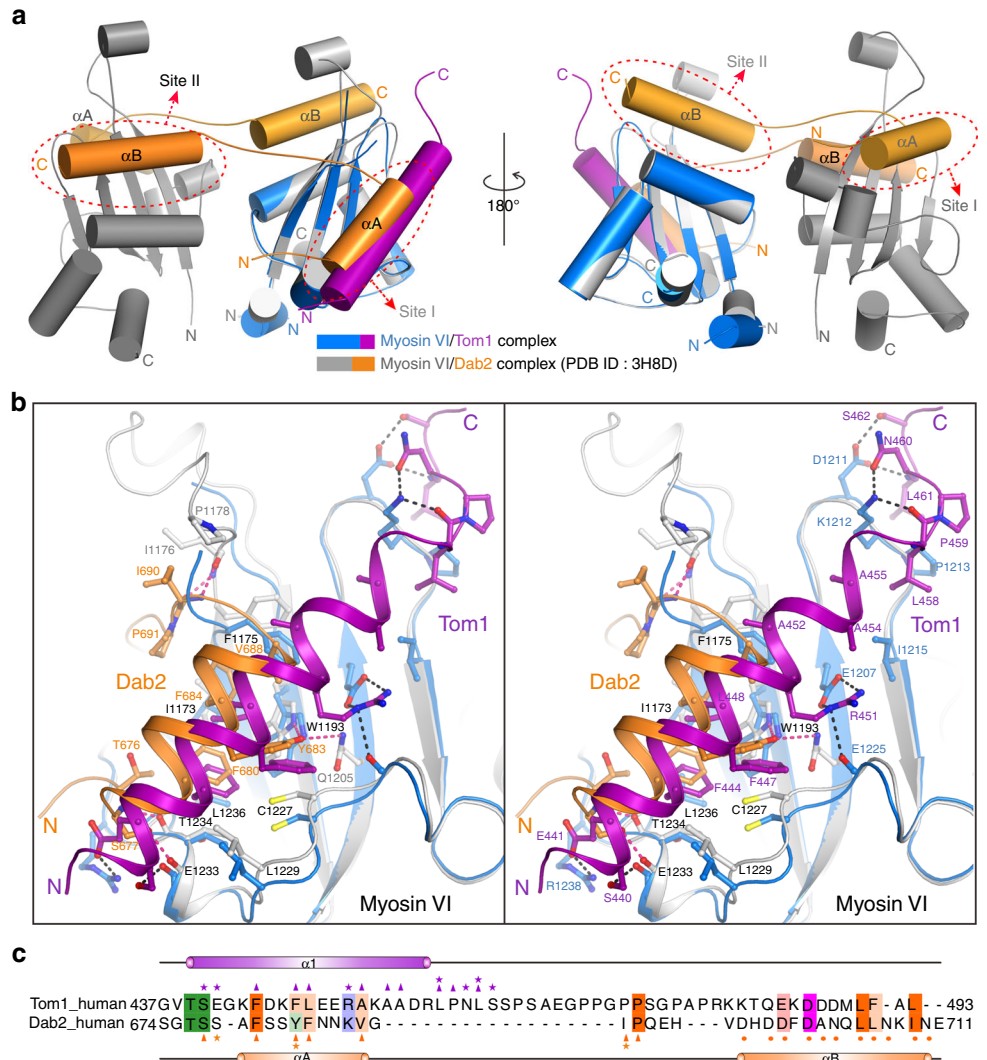

**Fig. 4** Comparisons of the Myosin VI/Tom1 and Myosin VI/Dab2 complexes. **a** The comparison of the overall structures of the Myosin VI/Dab2 complex (gray-orange, PDB ID: 3H8D) and the Myosin VI/Tom1 complex (blue-magenta). In this presentation, the positions of site I and site II in the C-terminal CBD of Myosin VI are further indicted. **b** Stereo view in the ribbon-stick model showing the comparison of the binding interfaces of the Myosin VI/Dab2 complex and the Myosin VI/Tom1 complex with the same color scheme as in **a**. The hydrogen bonds and salt bridges involved in the interactions are shown as dotted lines. The binding interface residues of Tom1 and Dab2 are labeled with magenta and orange numbers, respectively. While, the interface residues of the C-terminal CBD of Myosin VI that involved in the interactions with both Tom1 and Dab2, only for the interaction with Tom1 or Dab2, are labeled with black, blue, and gray numbers, respectively. **c** Structure-based sequence alignment of Tom1(437–493) and the Myosin VI-binding region of Dab2. In this structure-based sequence alignment, the conserved hydrophobic residues, polar neutral residues, positively charged residues and negatively charged residues are colored in orange, green, blue, and magenta, respectively. Key interface residues of Tom1 involved in the interaction with the site I of the C-terminal CBD of Myosin VI through the polar interactions and the hydrophobic interactions are further labeled with magenta stars and triangles, respectively, and that of Dab2 are highlighted with orange stars and triangles. Meanwhile, key interface residues of Dab2 that are critical for binding to the site II of the C-terminal CBD of Myosin VI, are labeled with orange dots

negligible chemical shift changes (Supplementary Fig. 7). These NMR results are well consistent with our aforementioned ITC results (Fig. 1c and Supplementary Fig. 3c, e). Therefore, the extreme C-terminal region of Tom1 is unable to directly interact with the C-terminal CBD of Myosin VI like that of the αB segment of Dab2. Taken together, based on these structural and sequence observations, we concluded that Myosin VI uses different binding modes to interact with Tom1 and Dab2, and importantly, the Myosin VI/Tom1 complex solved in this study represents a unique cargo recognition mode of Myosin VI.

**The cellular interactions of Myosin VI and Tom1 variants.** Tom1 was known to associate with Myosin VI in transfected

cells[12]. Therefore, we used co-immunoprecipitation (Co-IP) assay to evaluate the role of the interaction between Tom1 MBM and the C-terminal CBD of Myosin VI on the cellular association of Tom1 and Myosin VI in transfected cells. When co-transfected, the Cherry-tagged Tom1 full length was co-immunoprecipitated well with the GFP-tagged Myosin VI(1060–1285), a longer Myosin VI fragment including both the RRL motif and the WWY motif of Myosin VI (Fig. 5a). However, when co-transfection of the GFP-tagged Myosin VI(1060–1285) with the Tom1 L448E or R451E mutant, each of which was proved to lose its ability to interact with Myosin VI in vitro (Supplementary Fig. 5c, d and Table 1), the Cherry-tagged Tom1 mutant was unable to undergo a co-immunoprecipitation with GFP-tagged Myosin VI (1060–1285) (Fig. 5a). Conversely, we also assayed the W1193A

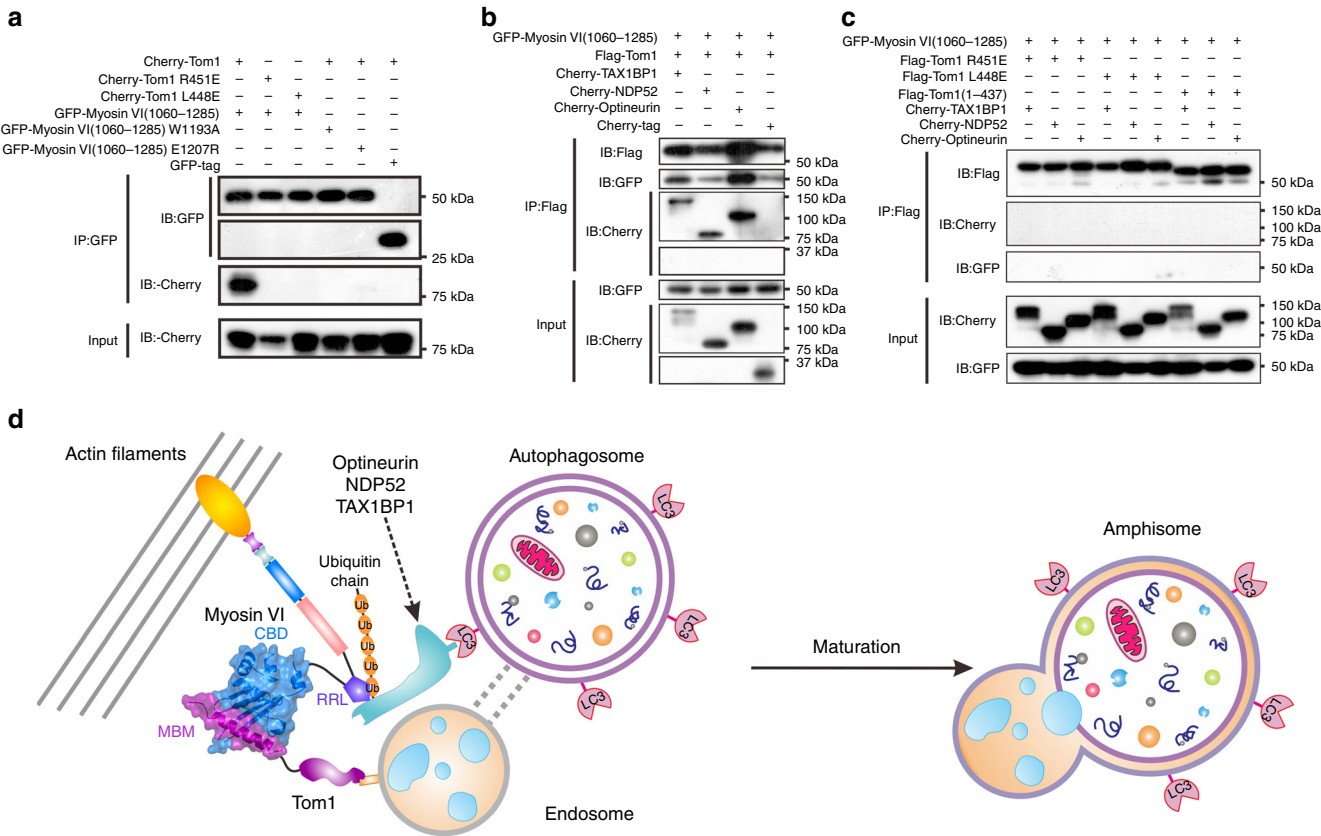

**Fig. 5** Myosin VI can link Tom1 with autophagy receptors. **a** A co-immunoprecipitation assay showing that point mutations of key interface residues observed in the Myosin VI/Tom1 complex structure abolish the specific interaction between Myosin VI(1060–1285) and Tom1 in cells. In this assay, cell extracts were prepared from HEK293T cells co-transfected with different combinations of plasmids as indicated, and 5% of each extracts were used as loading controls (bottom panel). **b** A co-immunoprecipitation assay revealing that Myosin VI(1060–1285), Tom1 and autophagy receptor TAX1BP1, NDP52, or Optineurin, can form ternary complexes in co-transfected cells. **c** A co-immunoprecipitation assay showing that mutations of Tom1, which can disrupt the interaction between Myosin VI(1060–1285) and Tom1, can also abolish the formation of the Tom1/Myosin VI/autophagy receptor ternary complex in cells. In this assay, cell extracts were prepared from HEK293T cells co-transfected with different combinations of plasmids as indicated, and 5% of each extracts were used as loading controls (bottom panel). Source data are provided as a Source Data file. **d** A proposed model depicting the tethering of endosome and autophagosome mediated by Myosin VI in cooperate with Tom1, the autophagy receptors, NDP52, TAX1BP1, and Optineurin as well as relevant ubiquitin chains, for facilitating the maturation of autophagosome in autophagy

and E1207R mutants of Myosin VI, as point mutations of these two residues of Myosin VI could completely abolish the interaction between Tom1 MBM and the C-terminal CBD of Myosin VI in our ITC-based assays (Supplementary Fig. 5a, b and Table 1). In line with our structural data (Fig. 3a–c), the W1193A and E1207R mutations of Myosin VI both essentially abolished the specific interaction between Tom1 and Myosin VI (1060–1285) in cells (Fig. 5a). Collectively, all these data clearly demonstrated that the specific interaction between Tom1 MBM and the C-terminal CBD of Myosin VI is essential for the association of Tom1 and Myosin VI in cells.

**Myosin VI links Tom1 with autophagy receptors**. In addition to Tom1, the C-terminal globular cargo-binding domain of Myosin VI was also implicated in the interactions with three autophagy receptors, NDP52, TAX1BP1, and Optineurin[12,28,33,42]. However, the detailed binding mechanism as well as the relationship between Tom1 and these autophagy receptors in binding to Myosin VI was still elusive. Therefore, we also sought to characterize the interactions of Myosin VI with NDP52, TAX1BP1, and Optineurin. Using Co-IP assays, we confirmed that NDP52, TAX1BP1 and Optineurin can specifically associate with Myosin VI(1060–1285) in cells (Supplementary Fig. 8a). Moreover, further NMR experiments using [15]N-labeled Myosin VI(1073–1119)

(a Myosin VI fragment only containing the RRL motif region without the C-terminal CBD) titrated with TAX1BP1(725–789) (a fragment including the C-terminal tandem zinc-fingers of TAX1BP1), NDP52(365–446) (a NDP52 fragment including its C-terminal two zinc-fingers), or Optineurin(417–512) (a fragment of Optineurin including the UBAN region), showed that a select set of peaks in the [1]H-[15]N HSQC spectrum of Myosin VI (1073–1119) underwent significant dose-dependent chemical shift changes or peak-broadenings (Supplementary Fig. 9a, c, e), indicating that this RRL fragment of Myosin VI can directly interact with NDP52, TAX1BP1 and Optineurin, in accord with several recent studies[34,42,46]. When plotting the amide backbone chemical shift changes as a function of the residue number and mapping shift differences onto a previously determined NMR structure of Myosin VI(1071–1122) (PDB ID: 2N10), which adopts a compact helical architecture consisting of two α-helixes, revealed that the C-terminal region of the second α-helix of Myosin VI(1071–1122) is the major binding interface of Myosin VI for interacting with NDP52, TAX1BP1, and Optineurin, as this region showed the largest chemical shift changes in the presence of NDP52, TAX1BP1, or Optineurin proteins (Supplementary Fig. 9b, d, f). However, detailed chemical shift mapping analyses revealed that the chemical shift changing patterns induced by TAX1BP1(725–789), NDP52(365–446) and

Optineurin(417–512), are different, suggesting that Myosin VI may adopt distinct key interface residues to interact with these three autophagy receptors (Supplementary Fig. 9b, d, f). Based on our biochemical and structural results, we inferred that the C-terminal globular cargo-binding domain of Myosin VI may simultaneously bind to Tom1 through the C-terminal CBD, and to autophagy receptor NDP52, TAX1BP1, or Optineurin through the RRL motif region. Indeed, further Co-IP assays showed that in the presence of Myosin VI(1060–1285), Tom1 can readily form a ternary complex with Myosin VI and autophagy receptor TAX1BP1, NDP52, or Optineurin (Fig. 5b), while Tom1 alone cannot directly associate with these autophagy receptors (Supplementary Fig. 8b). Importantly, in contrast to the wild type Tom1, the Tom1 truncation mutant (residues 1–437) lacking the Myosin VI-binding region as well as the Tom1 L448E, R451E mutants, all of which lost their abilities to interact with Myosin VI (Fig. 5a, c), are unable to associate with TAX1BP1, NDP52, and Optineurin even in the presence of Myosin VI (Fig. 5c). Consistently, further endogenous Co-IP assays revealed that Myosin VI can associate with Tom1 and autophagy receptor TAX1BP1, Optineurin, or NDP52, to form different ternary complexes at endogenous levels in vivo (Supplementary Fig. 10). Accordingly, all these data clearly demonstrated that Myosin VI can function as a bridging adaptor to simultaneously interact with Tom1 and autophagy receptor TAX1BP1, NDP52 or Optineurin, forming ternary complexes.

## Discussion

Previous studies have well established that Myosin VI can exist both in nonprocessive monomer form and processive dimer or oligomer form in cells[17,18,23,47]. As a monomer, Myosin VI could perform load-dependent anchoring or tethering functions, while as a dimer or oligomer, it could move processively along actin filaments to mediate cargo trafficking and sorting. The switching between a monomer state and a dimer or oligomer state likely depends on specific cellular contexts, and grants Myosin VI with greater functional diversities in cells. Recently, increasing evidences showed that the monomer-dimer or monomer-oligomer transitions of Myosin VI are regulated by cargo adaptor proteins such as Dab2 and GIPC[21,23,36,38]. Particularly, two previous structural studies uncovered that the binding of Dab2 to the C-terminal CBD of Myosin VI and the interaction of GIPC with the Myosin VI RRL motif region can induce the cargo-mediated dimerization and oligomerization of Myosin VI, respectively[21,38]. Surprisingly, in this study, we found that although Tom1 can directly interact with the C-terminal CBD of Myosin VI, it adopts a unique binding mode to interact with the C-terminal CBD of Myosin VI, unlike that of Dab2 (Fig. 4a, b), and, more importantly, the binding of Tom1 by the C-terminal CBD of Myosin VI cannot induce the dimerization or oligomerization of Myosin VI C-terminal CBD (Figs. 1f and 4a). Although we cannot formally exclude the possibility that additional dimerization or oligomerization domains may exist in the full-length Myosin VI and/or Tom1, our data suggest that Myosin VI may stay as a nonprocessive monomer when binding to Tom1, in line with its functional role to tether endosome and autophagosome for promoting the autophagosome maturation in autophagy. Given that the Myosin VI/Dab2 and Myosin VI/GIPC complexes are the only currently known complex structures related to the cargo-bindings of Myosin VI, therefore the Myosin VI/Tom1 complex structure determined in this study may represent the atomic structure showing how a monomeric Myosin VI to associate with cargo proteins for performing an anchoring or tethering role. It is also worthwhile to mention that based on our structural and sequence analyses (Fig. 3), the Tom1 family proteins, including

Tom1, TomL1, and TomL2, are predicated to share a similar binding mode to interact with Myosin VI. In the future, it will be interesting to know the detailed relationship as well as the potential different functions of these Tom1 family proteins in binding to Myosin VI.

Interestingly, our structural data revealed that Myosin VI adopts different approaches to interact with Tom1 and Dab2 (Fig. 4a), but several interface residues in the site I of the C-terminal CBD of Myosin VI are overlapped in binding to Tom1 and Dab2 (Fig. 4b and Supplementary Fig. 1a). Apparently, Tom1 and Dab2 should compete with each other for binding to the C-terminal CBD of Myosin VI, thereby forming distinct complexes for different intracellular processes. Thus, our work may also provide a mechanistic explanation to the different cellular functions mediated by Dab2 and Tom1 in binding to Myosin VI. In addition, based on our NMR titration results together with the determined GIPC/Myosin VI complex structure (PDB ID: 5V6H), we inferred that GIPC and the autophagy receptors, TAX1BP1, NDP52, and Optineurin, should be mutual exclusive in binding to Myosin VI, as the regions in the Myosin VI RRL motif with significant backbone amide chemical shift changes when binding to autophagy receptor TAX1BP1, NDP52, or Optineurin revealed by our NMR analyses (Supplementary Fig. 9), are highly overlapped with the GIPC-binding interfaces of the Myosin VI RRL motif in the determined GIPC/Myosin VI complex structure (Supplementary Fig. 11). Unfortunately, although we could detect the direct interactions between Myosin VI(1073–1119) and the autophagy receptors, TAX1BP1, NDP52, and Optineurin, using the NMR spectroscopy (Supplementary Fig. 9), we were unable to obtain stable complex proteins for further biochemical and structural characterizations, as Myosin VI(1073–1119) and these autophagy receptor proteins were eluted separately on the size-exclusion column during the protein purification procedure. Meanwhile, after numerous trials, we failed to obtain crystals by directly mixing together Myosin VI and autophagy receptor proteins at an equimolar ratio for crystallization screens, likely due to the very weak interactions between Myosin VI and these autophagy receptor proteins. Nevertheless, additional studies are required to elucidate the precise binding mechanisms of Myosin VI with these autophagy receptors in the future.

In this study, our in vitro biochemical results showed that the interactions between the RRL motif of Myosin VI and autophagy receptors, TAX1BP1, NDP52, and Optineurin, are very weak. It is puzzling how Myosin VI may tightly associate with these autophagy receptors to fulfill a tethering role with autophagesome in vivo. Interestingly, a previous study demonstrated that the RRL motif region of Myosin VI can also specifically interact with ubiquitin proteins, and importantly, the ubiquitin binding by the RRL motif region of Myosin VI contributes to the Myosin VI/Optineurin interaction[34]. Consistently, our NMR experiment showed that the RRL motif region of Myosin VI can directly bind to mono-ubiquitin (Supplementary Fig. 12). In addition, further biochemical assays based on analytical gel-filtration chromatography coupled with sodium dodecyl sulfate polyacrylamide gel electrophoresis (SDS-PAGE) analysis using purified proteins, revealed that in the presence of K63-linked, or M1-linked di-ubiquitin proteins, TAX1BP1(725–789) and NDP52(365–446) can readily associate with Myosin VI(1060–1285) and Tom1 (437–463) to form quaternary complexes (Supplementary Fig. 13a–d). Strikingly, we do observe that K63-linked and M1-linked di-ubiquitin proteins can promote the association between Optineurin(417–512) and Myosin VI(1060–1285) (Supplementary Fig. 13e, f), but the effects are much weaker comparing with that of TAX1BP1(725–789) and NDP52(365–446) (Supplementary Fig. 13), presumably due to the unique ubiquitin-binding

ability of Optineurin(417–512), which was proved to selectively recognize M1-linked ubiquitin chains and very weakly bind to K63-linked ubiquitin chains, but is unable to interact with mono-ubiquitin[48]. By contrast, both TAX1BP1(725–789) and NDP52 (365–446) can through their secondary $C_2H_2$-type zinc-finger domains to indiscriminately recognize mono-ubiquitin, M1-linked and K63-linked ubiquitin chains[43,44]. Nevertheless, all these data together with other peoples' reports suggested that the strong associations of autophagy receptors TAX1BP1, NDP52, and Optineurin with Myosin VI in cells might involve ubiquitin chains, although the origins of the involved ubiquitin chains remain elusive. Interestingly, previous works showed that Optineurin can undergo ubiquitination, and the ubiquitinated form of Optineurin can strongly associate with Myosin VI[34,49]. Therefore, for Optineurin, the attached ubiquitin chains may directly promote its association with Myosin VI, but for TAX1BP1 and NDP52, it is still unknown whether they would be directly modified by ubiquitination. Apparently, more works are needed to elucidate the working mode and the precise function of ubiquitination for promoting the associations of these autophagy receptors with Myosin VI in cells.

So far, dozens of autophagy receptors have been identified to play critical roles in different selective autophagy processes, but only TAX1BP1, NDP52 and Optineurin have been demonstrated to have the abilities to interact with Myosin VI[12,15,22,27,50,51]. Other people's reports together with our previous studies had well demonstrated that these Myosin VI-binding autophagy receptors have dual functions in autophagy, owning to their unique mutually exclusive abilities to interact with Myosin VI and ubiquitin[42–44,52]. Relying on their ubiquitin-binding abilities, they can recognize and target relevant ubiquitin-decorated autophagic cargoes into autophagosome for subsequent degradation. Meanwhile, they can also locate at the outside surface of autophagosome to associate with Myosin VI for promoting the maturation of autophagosome. Given that TAX1BP1, NDP52 and Optineurin, are likely competitive in binding to Myosin VI, whether they play redundant roles in promoting the maturation of autophagosome in different types of cells, and whether they could cooperate with Myosin VI to participate in selective autophagy processes mediated by other autophagy receptors that cannot directly bind to Myosin VI, still await further investigation.

Finally, based on our data together with previous other peoples' studies[12,15,22,42], we proposed a structure-based model depicting the tethering of endosome and autophagosome mediated by Myosin VI together with Tom1, autophagy receptors and relevant ubiquitin chains, for facilitating the maturation of autophagosome in autophagy (Fig. 5d). In this model, Myosin VI functioned as a nonprocessive monomer, and was anchored to the actin cytoskeleton through its N-terminal motor domain. While, the C-terminal part of Myosin VI associated with Tom1 mediated by the specific interaction between the C-terminal CBD of Myosin VI and the MBM of Tom1 (Fig. 5d), and meanwhile, through its RRL motif region, Myosin VI could further strongly associate with the autophagy receptor Optineurin, NDP52, or TAX1BP1 located at the outside surface of the autophagosome mediated by relevant ubiquitin chains, thereby forming a unique complex to bring endosome and autophagosome into close contact for the subsequent fusion to form amphisome (Fig. 5d).

## Methods

**Materials**. HEK293T (CBTCCAS, GNHu17) and HeLa cell (CBTCCAS, TCHu187) lines were kindly provided by Prof. Junying Yuan from Interdisciplinary Research Center on Biology and Chemistry, CAS, Shanghai, China. The full-length human Myosin VI plasmid was a gift from Prof. Mingjie Zhang from Hong Kong University of Science and Technology, Hong Kong, China. The full-length human Tom1, Optineurin, NDP52, TAX1BP1, and ubiquitin plasmids were obtained from Prof. Jiahuai Han from School of Life Sciences, Xiamen University, Xiamen, China.

**Protein expression and purification**. The different DNA fragments encoding human Tom1 (residues 215–493, 215–392, 392–493, 392–463, 392–437, 437–463, 437–493, 464–493), human Myosin VI (residues 1032–1285, 1032–1157, 1073–1119, 1157–1285, 1060–1285) were cloned into the pET-32M-3C vector (a modified version of pET32a vector containing a N-terminal thioredoxin-tag and His6-tag) or the pRSF-Trx vector for recombinant protein expressions (Supplementary Table 1). The plasmids of human TAX1BP1(725–789), human NDP52 (365–446) and mouse Optineurin(417–512) were prepared from our previous studies[43,44,48]. For fluorescence imaging experiment, the DNA fragments encoding human Tom1 were cloned into pmCherry-C1 vector or Flag vector, Myosin VI (1060–1285) were cloned into pEGFP-C1 vector, while NDP52, TAX1BP1, Optineurin were cloned into pmCherry-C1 (Supplementary Table 1). All the point mutations of Tom1 and Myosin VI used in this study were created using the standard PCR-based mutagenesis method, further checked by PCR screen using Taq Master mix (Vazyme Biotech Co., Ltd.) enzyme and confirmed by DNA sequencing.

Recombinant proteins were expressed in BL21 (DE3) *E. coli* cells (Invitrogen) induced by 100 μM IPTG at 16 °C. The bacterial cell pellets were re-suspended in the binding buffer (50 mM Tris, 500 mM NaCl, 5 mM imidazole at pH 7.9), and then lysed by the ultrahigh pressure homogenizer FB-110XNANO homogenizer machine (Shanghai Litu Machinery Equipment Engineering Co., Ltd.). Then the lysis was spun down by centrifuge at $35,000 \times g$ for 30 min to remove the pellets fractions. His6-tagged proteins were purified by $Ni^{2+}$-NTA agarose (GE Healthcare) affinity chromatography, while GST-tagged proteins were purified by glutathione sepharose 4B (GE Healthcare) affinity chromatography. Each recombinant protein was further purified by size-exclusion chromatography. The N-terminal tag of each recombinant protein was cleaved by 3C protease, and further removed by size-exclusion chromatography. Uniformly $^{15}N$ or $^{15}N/^{13}C$-labeled Tom1 or Myosin VI fragment proteins were prepared by growing bacteria in M9 minimal medium using $^{15}NH_4Cl$ (Cambridge Isotope Laboratories Inc.) as the sole nitrogen source or $^{15}NH_4Cl$ and $^{13}C_6$-glucose (Cambridge Isotope Laboratories Inc.) as the sole nitrogen and carbon sources, respectively.

**Preparations of relevant di-ubiquitin proteins**. The K63R-Ub and Ub-D77 mono-ubiquitin mutants were cloned into the pET-M-3C vector (a modified version of the pET32a vector containing a N-terminal His6-tag), and expressed and purified as mono-ubiquitin for enzymatic synthesis of K63-linked di-ubiquitin according to published protocols[53,54]. For M1-linked di-ubiquitin, two repeats of a DNA fragment encoding the human ubiquitin (residues 1–76) were directly fused together as one extended DNA fragment, which was cloned into pET-32M-3C vectors. The M1-linked di-ubiquitin proteins were expressed and purified following the same procedure for mono-ubiquitin.

**Analytical gel-filtration chromatography**. Purified proteins were loaded on to a Superose 200 increase 10/300 GL column (GE Healthcare) equilibrated with a buffer containing 20 mM Tris-HCl (pH 7.9), 100 mM NaCl and 1 mM DTT. Analytical gel-filtration chromatography was carried out on an AKTA FPLC system (GE Healthcare). The fitting results were further output to the Origin 8.5 software and aligned with each other.

**NMR spectroscopy**. The $^{15}N$-labeled protein samples for NMR titration experiments were concentrated to ~0.1 mM, except for Myosin VI(1157–1285), which can be only concentrated to 0.05 mM, and the $^{15}N/^{13}C$-labeled protein samples were concentrated to ~0.6 mM for backbone resonance assignment experiments. All the protein samples for NMR studies are in the 50 mM potassium phosphate buffer containing 100 mM NaCl, and 1 mM DTT at pH 6.5, and NMR spectra were acquired at 25 °C or 30 °C on an Agilent 800 MHz spectrometer equipped with an actively z gradient shielded triple resonance cryogenic probe at the Shanghai Institute of Organic Chemistry. Backbone resonance assignments of the Myosin VI (1073–1119) and Tom1(392–493) fragments were achieved using a suite of heteronuclear correlation experiments, including HNCO, HNCACB, and CBCA(CO)NH using a $^{15}N/^{13}C$-labeled protein samples[55].

**Isothermal titration calorimetry (ITC) assay**. ITC measurements were carried out on an ITC200 (GE Healthcare) or Microcal PEAQ-ITC (Malvern) calorimeter at 25 °C. All protein samples were in the same buffer containing 20 mM Tris (pH 7.9), 100 mM NaCl, and 1 mM DTT. The concentrated 50 μM proteins were loaded into the cell and 500 μM of proteins were loaded into the syringe, respectively. The titration processes were performed by injecting 40 μl aliquots of the proteins in syringe into cell at time intervals of 120 or 150 s to ensure that the titration peak returned to the baseline. The titration data were analyzed using the program Origin 8.5 from Micro Cal and fitted using the one-site binding model.

**Multi-angle light scattering**. For multi-angle light-scattering measurement, Myosin VI(1157–1285) protein, Tom1(437–463) protein, and Myosin VI (1157–1285)/Tom1(437–463) complex samples were injected into an AKTA FPLC system (GE Healthcare) with a Superose 200 increase 10/300 GL column (GE Healthcare) with the column buffer containing 20 mM Tris-HCl, pH 7.9, 100 mM NaCl, 1 mM DTT. The chromatography system was coupled to a static light-scattering detector (miniDawn, Wyatt Technology) and a differential refractive

index detector (Optilab, Wyatt Technology). Data were collected every 0.5 s with a flow rate of 0.5 ml/min. Data were analyzed using the ASTRA 6 software (Wyatt Technology) and drawn by the Origin 8.5 software.

**Analytical ultracentrifugation.** Sedimentation velocity experiments were performed on a Beckman XL-I analytical ultracentrifuge equipped with an eight-cell rotor under $142,250 \times g$ at 20 °C. The partial specific volume of different protein samples and the buffer density were calculated using the program SEDNTERP (http://www.rasmb.bbri.org/). The final sedimentation velocity data were analyzed and fitted to a continuous sedimentation coefficient distribution model using the program SEDFIT[56]. The fitting results were further output to the Origin 8.5 software and aligned with each other.

**Protein crystallization and structural elucidation.** Crystals of Tom1(437–463)/ Myosin VI(1157–1285) complex were obtained by mixing the freshly purified complex proteins (10 or 20 mg/ml in 20 mM Tris-HCl, 100 mM NaCl, 1 mM DTT at pH 7.9) with equal volumes of reservoir solution containing 0.2 M sodium acetate trihydrate, 0.1 M TRIS hydrochloride pH 8.5, 30% w/v polyethylene glycol 4000 using the sitting-drop vapor-diffusion method at 16 °C. Before diffraction experiments, glycerol (10%) was added as the cryo-protectant. A 1.8 Å resolution X-ray data set for Tom1(437–463)/Myosin VI(1157–1285) complex were collected at the beamline BL19U1 of the Shanghai Synchrotron Radiation Facility[57]. The diffraction data were processed and scaled using HKL2000[58].

The phase problem of Tom1(437–463)/Myosin VI(1157–1285) complex was solved by molecular replacement method using the modified Myosin VI structure in the Myosin VI/Dab2 complex (PDB ID: 3H8D) as the search model with PHASER[59]. The initial structural model was rebuilt manually using COOT[60], and then refined using REFMAC[61], or PHENIX[62]. Further manual model building and adjustment were completed using COOT[60]. The qualities of the final model were validated by MolProbity[63]. The final refinement statistics of solved structures in this study were listed in Table 2. All the structural diagrams were prepared using the program PyMOL (http://www.pymol.org/).

**Co-immunoprecipitation.** HEK293T cells or transfected HEK293T cells were lysed in ice-cold cell lysis buffer (50 mM Tris, pH 7.9, 150 mM NaCl, 0.5% Nonidet P-40, 1 mM phenylmethylsulfonyl fluoride, 1% protease inhibitor cocktail) for 1 h at 4 °C, and followed by centrifugation at $16,873 \times g$ for 15 min at 4 °C. For co-immunoprecipitation using transfected cells, the supernatant fraction of the transfected HEK293T cells was then incubated with anti-GFP conjugated agarose beads (Medical & Biological Laboratories) for 4 h or anti-Flag conjugated agarose beads (Sigma-Aldrich) for 2 h at 4 °C, respectively. The beads were washed with the cell lysis buffer for five times and re-suspended with the SDS-PAGE sample buffer. The prepared samples were separated by 10% SDS-PAGE and analyzed using western blot. For endogenous immunoprecipitation, the supernatant fraction of the HEK293T was then incubated with an antibody to the tail region of Myosin VI (SANTA CRUZ, catalog no. sc-393558, 1:100 dilutions) for 1 h and then bound to Protein G agarose beads (Invitrogen) for another hour. The beads were washed with the ice-cold cell lysis buffer for five times and re-suspended with the SDS-PAGE sample buffer. The prepared samples were separated by 4–15% gradient gels, blotted, and analyzed using antibodies to Myosin VI (proteintech, catalog no. 26778-1-AP, 1:1000 dilutions), Tom1 (Abcam, catalog no. ab99356, 1:1000 dilutions), TAX1BP1 (Abcam, catalog no. ab245636, 1:2000 dilutions), NDP52 (Abcam, catalog no. ab68588, 1:1000 dilutions) and Optineurin (Abcam, catalog no. ab213556, 1:1000 dilutions), respectively. Control immunoglobulin G (IgG) immunoprecipitation was performed using a normal mouse IgG (SANTA CRUZ, catalog no. sc-2025, 1:100 dilutions).

**Reporting summary.** Further information on research design is available in the Nature Research Reporting Summary linked to this article.

## Data availability

The coordinate and structure factor of the C-terminal CBD of Myosin VI and Tom1 MBM complex has been deposited in the Protein Data Bank with accession number 6J56. Data supporting the findings of this manuscript are available from the corresponding author upon reasonable request. A reporting summary for this Article is available as a Supplementary Information file. The source data underlying Figs. 1c–f, 5a–c, and Supplementary Figs. 2a–f, 3a–g, 5a–d, 7b, 8a, b, 9b, d, f, 10a–c, 12b, 13a–f are provided as a Source Data file.

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

## Acknowledgements

We thank SSRF BL19U1 for X-ray beam time, Dr. Jianchao Li for help in the X-ray diffraction data collection, Prof. Jiahuai Han for the full-length Tom1, Optineurin, NDP52, TAX1BP1 cDNA, Prof. Mingjie Zhang for the human Myosin VI plasmid. This work was supported by grants from the National Natural Science Foundation of China (21822705, 31470749, 21621002), the National Key R&D Program of China (2016YFA0501903), the Science and Technology Commission of Shanghai Municipality (17JC1405200), the Strategic Priority Research Program of the Chinese Academy of Sciences (XDB20000000), the start-up fund from State Key Laboratory of Bioorganic and Natural Products Chemistry and Chinese Academy of Sciences (for L.P.); a grant from the National Natural Science Foundation of China (31800646) (for Yingli W.).

## Author contributions

S.H. and L.P. designed research; S.H., Y.G., Yingli W., Y.L., F.T., Z.Z., Yaru W. and J.L. performed research; S.H., Y.G. and L.P. analyzed data; S.H. and L.P. wrote the paper.

## Additional information

**Competing interests:** The authors declare no competing interests.

