## [Peer Review File · Nature Communications]

Reviewers' Comments:

Reviewer #1:

Remarks to the Author:

This manuscript investigates the mechanism of myosin VI interaction with the endosomal protein Tom1. The strength of this manuscript is the rigor used to provide atomic level insights into the nature of myosin VI/Tom1 interaction. Biochemical methods were used to define the myosin VI/Tom1 interaction domains with ITC providing binding constants, followed by x-ray crystallography to solve the structure of the binding domains in complex. Structure-based mutations were made to demonstrate that interaction of the expressed proteins in cells requires contacts revealed by the structure. Overall this manuscript is quite strong, but some concerns should be addressed that are raised below.

1. Fig 2 and the section around line 194: alpha2 does not appear to be a true helix and even alpha3 is questionable. How many amino acids do these helices span? Also amino acid numbers should be included in (a) particularly at the terminal ends. Sub-fig (b) is not very informative.
2. The clash overlap for the deposited structure raise concerns particularly those involving backbone atoms.
3. Figure 3: Inclusion of myosin VI amino acid labels in (a) and (b) would be helpful. What is the basis of the color coding in (d)? Toning it down to highlight main points may be more effective. This concern also applies to Figure 4c.
4. Figure 6: The authors demonstrate that Tom1 cannot be immunoprecipitated with optineurin/NDP53/TAX1BP1 unless myosin VI is present. However, it is not clear whether the interaction of Tom1 with these proteins is dependent on protein over-expression. In particular, does this ternary complex occur at endogenous levels?

Overall the manuscript is written clearly. However it does need a read over by a native English speaker. For example, on line 78 and 98 'server' and 'servers' should be 'serve' and 'serves'.

Reviewer #2:

Remarks to the Author:

In this study Hu et al present the first crystal structure of myosin VI in complex with its known adaptor protein Tom1. In contrast to previous structures of the myosin VI/GIPC or the myosin VI/Dab2 complex, the binding of a 30 amino acid fragment of Tom1 does not induce dimerization of the myosin VI cargo binding domain. The high-resolution structure shown in this paper demonstrated that the C-terminal region of the cargo binding domain (CBD) of myosin VI forms a 1:1 complex with this short Tom1 fragment. The crystallographic data is informative and well-presented and makes an important contribution to our knowledge of cargo binding by this myosin motor. However, as outlined below, I have a number of important questions and suggestions regarding the biological data presented, which should be addressed before publication.

1. The 'monomeric cargo recognition mode' has not been demonstrated experimentally. The authors show that a short peptide of Tom1 (437-463) and the part of the myosin VI CBD (1032-1285) are monomeric, however, dimerization domains may exist in the full-length proteins. Therefore, these data have to be discussed carefully and should not be included in the title.
2. Please modify in the introduction in the description of the domain organisation of myosin VI (line 84-89): important missing domains are the reverse gear in the neck region and the 3-helix bundle region before the SAH domain. In addition the cargo-binding domain (CBD) contains both the N-terminal RRL and the C-terminal WWY motif (1060-1285). In the manuscript the CBD only contains the WWY, which is not correct. This should also be corrected in the schematic in figure

1a.

3. Please correct the use of the myosin VI CBD throughout the text accordingly.

4. The alignment highlighting the conserved IEXWL motif of Tom1 (line 161) is shown in Arden et al., Biochemical Journal 2016.

5. The discussion should also include the paper by He et al. Cell Rep. 2016, which demonstrates that binding of optineurin to myosin VI requires a ubiquitin-binding domain overlapping with the RRL motif. This suggests that higher affinity binding of optineurin and the other autophagy receptors to myosin VI in cells might involve ubiquitin chains, which would explain lower affinity complexes of the purified proteins in vitro.

Figure 1: Please correct the position of the CBD to include the RRL motif.

Explain and correct the following abbreviations in the figure legend: SAH single alpha helix region, UBAN ?, SKICH ?,

Figure 4: Please indicate the position of site I and site II

Figure 5: The immunofluorescence localisation of the Myosin VI CBD (1060-1285) and Tom1 shown in figure 5 is not convincing. Both proteins should localise to numerous APPL1/Rab5 positive endosomes in the cell periphery. However, the images in figure 5 show large punctae distributed throughout the cell concentrating in the perinuclear region, possibly cellular protein aggregates. These experiments should be repeated to show the correct localisation of Tom1 and Myosin VI to APPL1 endosomes in the cortical actin network. The biochemical characterisation of the mutant proteins shown in figure 5g is far more convincing and it raises the question whether the experiments shown in figure 5 a-f are required. For the statistical analysis in figure 5f an Anova test and not a student t-test should be used.

Figure 6. Most of the results shown in this figure including the model are not novel. Figure 6a: Coimmunoprecipitation of myosin VI and the autophagy receptors has been published previously. Figure 6b: Simultaneous binding of Tom1 and NDP52 to the CBD of myosin VI was previously shown in Arden et al. Biochemical Journal, 2016; Figure 5F. Finally, a very similar model was proposed in Fig. 4C in Tumbarello et al. JCS 2013 highlighting that myosin VI can act as a tether that links endosomes via Tom1 and the autophagy adaptors to autophagosomes.

Minor points

Please correct:

line 83 exocytosis, delete at the Golgi complexes

line 85 that can bind to and walk along actin filaments

line 98 serves

108 was proved to be

109 receptors

110 which are

114 endosomes

115 autophagosomes

121 family members, which can ...

121interact with the C-terminal CBD of Myosin VI

123that the C-terminal CBD of Myosin VI adopts a unique ...

127 receptors

150 ...rather than the N-terminal "RRL" motif region (residues 1032-1157) of the Myosin VI CBD is required ...

384 as their binding sites in the mMyosin VI "RRL" motif are highly overlapped. Not clear, please explain.

Further corrections of the English are required.

Reviewer #3:

Remarks to the Author:

In this manuscript, Hu et al investigated the interaction between the Myosin VI cargo binding domain (CBD) and Tom1, a component of the endosome sorting complex. They discovered, by detailed biochemical and structural characterization, that Myosin VI CBD binds to a short and continuous helix from Tom1. Interestingly, they provided evidences showing that Myosin VI CBD and Tom1 form a 1:1 stoichiometric complex, instead of a 2:2 complex previously characterized for other Myosin VI CBD targets such as Dab2. Formation of the 1:1 stoichiometric complex between Myosin VI CBD and Tom1 suggests that Myosin VI likely functions as a tether between actin filaments and endosomes via binding to Tom1. The authors further showed that the "RRL"-motif of Myosin VI preceding its CBD binds to autophagy receptors like Optineurin, NDP52 and TAX1BP1. With these findings, the authors proposed a model suggesting that Myosin VI tail cargo binding domains can simultaneously tether autophagosome and endosome and promote formation of amphisome. This study provides direct experimental evidence showing that the tail cargo binding domains of Myosin VI may determine the molecular tethering role of the motor via forming a 1:1 heterodimer with its cargo proteins. Therefore, the manuscript is interesting for understanding cellular functions of Myosin VI and also other unconventional myosins. As such, the current manuscript is a strong candidate for NC.

However, there are a number of issues will need to be addressed before the manuscript can be accepted for publication:

1. The evidence that the regions outside aa 437-463 are not involved in binding to Myosin VI CBD is not very strong. The heat releases shown in Fig. 1C&E are significantly different, suggesting that residues N-terminal to 437-463 may also be involved. The author should show both entropy and enthalpy values of their ITC titration curves, as these values will provide evidence whether regions outside 437-463 may also be involved in binding.
2. The NMR data shown in Figure S6 indicated that residues C-terminal to 437-463 are involved in binding to CDB as obvious chemical shift differences are observed. The NMR data indicate that this binding is rather weak but may be important.
3. It will probably be more sensitive to titrate unlabelled Myosin VI CBD to ¹⁵N-labeled Tom1 392-493 (or other fragments if more suitable) to investigate whether residues outside aa 437-463 may be involved in Myosin VI binding by careful NMR-based investigations.
4. The authors have identified single point mutations in Tom1 which can abolish the Myosin VI/Tom1 interaction. It will add a lot value to the current manuscript if the authors can demonstrate that such Tom1 mutants are defective in amphisome formation as proposed by the authors.
5. The authors would need to explain how the extremely weak bindings between the Myosin VI "RRL"-motif with autophagy receptors may fulfil the motor's tethering role with autophagosome.

Point-by-point responses to the reviewers' comments:

(Reviewers' comments are in **blue**, and our responses are in **black**)

Specific responses to the criticisms from the Reviewer #1:

We are very grateful to the enthusiastic comments and insightful suggestions from the reviewer. The following is our detailed responses to the reviewer's comments:

This manuscript investigates the mechanism of myosin VI interaction with the endosomal protein Tom1. The strength of this manuscript is the rigor used to provide atomic level insights into the nature of myosin VI/Tom1 interaction. Biochemical methods were used to define the myosin VI/Tom1 interaction domains with ITC providing binding constants, followed by x-ray crystallography to solve the structure of the binding domains in complex. Structure-based mutations were made to demonstrate that interaction of the expressed proteins in cells requires contacts revealed by the structure. Overall this manuscript is quite strong, but some concerns should be addressed that are raised below.

1. Fig 2 and the section around line 194: alpha2 does not appear to be a true helix and even alpha3 is questionable. How many amino acids do these helices span? Also amino acid numbers should be included in (a) particularly at the terminal ends. Sub-fig (b) is not very informative.

We thank reviewer for pointing out this for us, and we agree with reviewer's comment that alpha2 is not a true helix. However, alpha3 is a real helix, which spans four amino acid residues (1230-1234). Consistently, the corresponding alpha3 region of Myosin VI in the reported Myosin VI/Dab2 structure is also assigned as a helix¹. Accordingly, we have removed the alpha2 and re-assigned the secondary structure feature of Myosin VI in all related structure and sequence alignment figures in the revised manuscript. In addition, following reviewer's comments, we have included the amino acid numbers at the terminal ends in panel (a) of Fig. 2 and further modified panel (b) to include an "open-book" view of the binding interface between Myosin VI and Tom1 (see **Figure I** below, and **Fig.2** in the revised manuscript).

Figure I: The overall structure of the C-terminal CBD of Myosin VI and Tom1 MBM complex. (a) Ribbon diagram showing the overall structure of the C-terminal CBD of Myosin VI and Tom1 MBM complex. In this drawing, the C-terminal CBD of Myosin VI is shown in blue, and Tom1 MBM in magenta. (b) Surface representations showing the overall architecture of Myosin VI/Tom1 complex (left panel), and the “open-book” view of the binding interface between Myosin VI and Tom1 (right panel) with the same color scheme as in panel a.

2. The clash overlap for the deposited structure raise concerns particularly those involving backbone atoms.

We are sorry that we have not carefully checked the clash overlap issue for our deposited structure. Following reviewer’s comment, we have taken extra care to improve the refinement of the deposited structure. The newly refined structure has no clash overlap involving backbone atoms. We have replaced the originally deposited PDB structure with the newly refined one.

target proteins. Our obtained results show that Myosin VI can associate with Tom1 and autophagy receptor TAX1BP1, Optineurin, or NDP52, to form different ternary complexes at endogenous levels (see **Figure III** below, and **Supplementary Fig. 10** in the revised manuscript). We have included this data in the revised manuscript.

Figure III: Myosin VI can associate with Tom1 and autophagy receptor TAX1BP1, Optineurin or NDP52 forming ternary complexes at endogenous levels. (a-c) Co-immunoprecipitation assays showing that Myosin VI can interact with Tom1 and autophagy receptor TAX1BP1 (a), Optineurin (b), or NDP52 (c) at endogenous levels *in vivo*. In this assay, cell extracts were prepared from HEK293T cells, and 5% of each extracts were used as loading controls (left panel). Immunoprecipitation was performed under native conditions using an antibody to the tail region of Myosin VI (central panel), then run out on a 4-15% gradient gel, blotted, and analyzed using antibodies to Myosin VI, Tom1, TAX1BP1, NDP52 and Optineurin, respectively. Control immunoglobulin G (IgG) immunoprecipitation (right panel) was performed using a normal mouse IgG.

Overall the manuscript is written clearly. However it does need a read over by a native English speaker. For example, on line 78 and 98 'server' and 'servers' should be 'serve' and 'serves'.

We thank the reviewer for pointing out these mistakes in our writing. We have corrected them in the revised manuscript. Furthermore, we have taken extra care to improve the writing of our manuscript.

Specific responses to the criticisms from the Reviewer #2:

We are very grateful to the enthusiastic comments and insightful suggestions from the reviewer. The following is our detailed responses to the reviewer's comments:

In this study Hu et al present the first crystal structure of myosin VI in complex with its known adaptor protein Tom1. In contrast to previous structures of the myosin VI/GIPC or the myosin VI/Dab2 complex, the binding of a 30 amino acid fragment of Tom1 does not induce dimerization of the myosin VI cargo binding domain. The high-resolution structure shown in this paper demonstrated that the C-terminal region of the cargo binding domain (CBD) of myosin VI forms a 1:1 complex with this short Tom1 fragment. The crystallographic data is informative and well-presented and makes an important contribution to our knowledge of cargo binding by this myosin motor. However, as outlined below, I have a number of important questions and suggestions regarding the biological data presented, which should be addressed before publication.

1. The 'monomeric cargo recognition mode' has not been demonstrated experimentally. The authors show that a short peptide of Tom1 (437-463) and the part of the myosin VI CBD (1032-1285) are monomeric, however, dimerization domains may exist in the full-length proteins. Therefore, these data have to be discussed carefully and should not be included in the title.

We agree with reviewer's comments that we have only studied parts of Tom1 and Myosin VI not the full-length proteins, and dimerization domains may exist in the full-length proteins. Following reviewer's nice suggestion, we have turned down our statements in the discussion section, and changed the title to "The structure of the Myosin VI and Tom1 complex reveals a unique cargo recognition mode of Myosin VI for tethering" in the revised manuscript.

2. Please modify in the introduction in the description of the domain organization of myosin VI (line 84-89): important missing domains are the reverse gear in the neck region and the 3-helix bundle region before the SAH domain. In addition the cargo-binding domain (CBD) contains both the N-terminal RRL and the C-terminal WWY motif (1060-1285). In the manuscript the CBD only contains the WWY, which is not correct. This should also be corrected in the schematic in figure 1a.

We are sorry for our carelessness and thank the reviewer for pointing out these mistakes in our writing. Following reviewer's comments, we have added the descriptions of the reverse gear as well as the 3-helix bundle region, and modified the description related to the domain organization of Myosin VI in the introduction section of the revised manuscript. In addition, we have also corrected the Figure 1a according to reviewer's suggestion (see **Figure IV** below, and **Fig. 1a** in the revised manuscript).

Figure IV: A schematic diagram showing the domain arrangements of Myosin VI, Tom1, NDP52, TAX1BP1 and Optineurin. In this drawing, domains involved in the protein-protein interaction are highlighted with black lines, and the relevant interactions between two proteins are indicated by two-way arrows.

3. Please correct the use of the myosin VI CBD throughout the text accordingly.

Following reviewer's suggestion, we have used the "C-terminal CBD" to replace the "CBD" used in the original manuscript, and we have corrected the use of the Myosin VI CBD throughout the text in the revised manuscript. Thanks.

4. The alignment highlighting the conserved IEXWL motif of Tom1 (line 161) is shown in Arden et al., Biochemical Journal 2016.

We are sorry for our careless reading and survey of the previously published papers. We have corrected this citation according to reviewer's suggestion in the revised manuscript.

5. The discussion should also include the paper by He et al. Cell Rep. 2016, which demonstrates that binding of optineurin to myosin VI requires a ubiquitin-binding domain overlapping with the RRL motif. This suggests that higher affinity binding of optineurin and the other autophagy receptors to myosin VI in cells might involve ubiquitin chains, which would explain lower affinity complexes of the purified proteins *in vitro*.

We thank the reviewer for this insightful suggestion. Indeed, we had not taken into account that ubiquitin chains can serve as a bridging adaptor to increase the binding affinity of autophagy receptors and Myosin VI in the original manuscript. Based on our *in vitro* biochemical results, the interactions between the "RRL" motif of Myosin VI and autophagy receptors, TAX1BP1, NDP52 and Optineurin, are extremely weak. It is

puzzling how Myosin VI may associate with these autophagy receptors to fulfill a tethering role with autophagosome *in vivo*. Following reviewer’s suggestion, we have performed a NMR titration experiment, which shows that the “RRL” motif region of Myosin VI can directly bind to mono-ubiquitin (see **Figure V** below, and **Supplementary Fig. 12** in the revised manuscript), consistent with the study by He and co-workers². Furthermore, our biochemical assays based on analytical gel filtration chromatography coupled with SDS-PAGE analysis using purified proteins, revealed that in the presence of K63-linked, or M1-linked di-ubiquitin proteins, TAX1BP1(725-789) and NDP52(365-446) can readily associate with Myosin VI(1060-1285) and Tom1(437-463) to form quaternary complexes (see **Figure VIa-d** below, and **Supplementary Fig. 13a-d** in the revised manuscript). Interestingly, we can observe that K63-linked and M1-linked di-ubiquitin proteins can promote the association between Optineurin(417-512) and Myosin VI(1060-1285) (see **Figure VIe-f** below, and **Supplementary Fig. 13e-f** in the revised manuscript), but the effects are much weaker comparing with that of TAX1BP1(725-789) and NDP52(365-446) (see **Figure VI** below, and **Supplementary Fig. 13** in the revised manuscript), presumably due to the unique ubiquitin-binding ability of Optineurin(417-512), which can selectively recognize M1-linked di-ubiquitin and very weakly bind to K63-linked di-ubiquitin, but is unable to interact with mono-ubiquitin³. By contrast, both TAX1BP1(725-789) and NDP52(365-446) can through their secondary C₂H₂-type zinc-finger domains to indiscriminately recognize mono-ubiquitin, M1-linked and K63-linked ubiquitin chains^{4,5}. Nevertheless, all these data suggest that ubiquitin chains can promote the strong associations of autophagy receptors TAX1BP1, NDP52 and Optineurin with Myosin VI *in vitro*, in line with the report by He and co-workers², which demonstrated that the ubiquitin binding by the “RRL” motif region of Myosin VI contributes to the Myosin VI/Optineurin interaction. Therefore, based on our data and other peoples’ reports, we infer that the strong associations of autophagy receptors TAX1BP1, NDP52 and Optineurin with Myosin VI in cells might involve ubiquitin chains. We have included these new data, and discussed them together with the findings from He and co-workers’ study in the discussion section of the revised manuscript.

Figure V: NMR-based characterizations of the interaction between Myosin VI(1073-1119) and mono-ubiquitin. (a) A superposition plot of the ^1H - ^{15}N HSQC spectra of Myosin VI(1073-1119) titrated with mono-ubiquitin at different molar ratios. (b) A plot of backbone amide chemical shift differences and peak-broadenings as a function of the residue number of Myosin VI(1073-1119) between the wild-type and the protein titrated with mono-ubiquitin at the molar ratio of 4:1. In this panel, the secondary structures and

amino acid sequence of Myosin VI(1073-1119) are also indicated at the top of the figure, and the insert shows the shift changes and peak-broadenings mapped onto a representative NMR structure of Myosin VI(1071-1122) (PDB ID: 2N10). In this representation, the residues with disappeared NMR peaks due to peak-broadenings are shown in blue, and the combined ^1H and ^{15}N chemical shift changes are defined as:

$$\Delta_{\text{ppm}} = [(\Delta\delta_{HN})^2 + (\Delta\delta_N \times \alpha_N)^2]^{1/2}$$

Where $\Delta\delta_{HN}$ and $\Delta\delta_N$ represent chemical shift differences of amide proton and nitrogen chemical shifts of the each residue of Myosin VI(1073-1119). The scaling factor (α_N) used to normalize the ^1H and ^{15}N chemical shift is 0.17.

Figure VI: K63-linked and M1-linked ubiquitin chains can promote the associations of autophagy receptors TAX1BP1, NDP52 and Optineurin with Myosin VI *in vitro*. (a and b) Analytic gel filtration chromatography coupled with SDS-PAGE analysis using purified proteins, revealed that in the presence of K63-linked di-ubiquitin (a), or M1-linked di-ubiquitin (b), TAX1BP1(725-789) can readily associate with Myosin VI(1060-1285) and Tom1(437-463) to form quaternary complexes. In each panel, the insert shows

the SDS-PAGE combined with Coomassie-blue staining analysis of the protein components of the indicated “Fraction I” fraction collected from the analytic gel filtration chromatography experiment of the Myosin VI(1060-1285)/Tom1(437-463)/di-ubiquitin/TAX1BP1(725-789) mixture (the black curve). (c and d) Analytic gel filtration chromatography coupled with SDS-PAGE analysis using purified proteins, revealed that in the presence of K63-linked di-ubiquitin (c), or M1-linked di-ubiquitin (d), NDP52(365-446) can readily associate with Myosin VI(1060-1285) and Tom1(437-463) to form quaternary complexes. In each panel, the insert shows the SDS-PAGE combined with Coomassie-blue staining analysis of the protein components of the indicated “Fraction I” fraction collected from the analytic gel filtration chromatography experiment of the Myosin VI(1060-1285)/Tom1(437-463)/di-ubiquitin/NDP52(365-446) mixture (the black curve). (e and f) Analytic gel filtration chromatography coupled with SDS-PAGE analysis using purified proteins, revealed that K63-linked di-ubiquitin (e), and M1-linked di-ubiquitin (f), can weakly promote the association of Optineurin(417-512) with Myosin VI(1060-1285). In each panel, the lower part shows the SDS-PAGE combined with Coomassie-blue staining analysis of the protein components of the indicated fractions collected from the analytic gel filtration chromatography experiment of the Myosin VI(1060-1285)/Tom1(437-463)/di-ubiquitin/Optineurin(417-512) mixture (the black curve).

Figure 1: Please correct the position of the CBD to include the RRL motif. Explain and correct the following abbreviations in the figure legend: SAH single alpha helix region, UBAN ?, SKICH ?

We thank the reviewer for pointing out these mistakes. We have corrected them in the updated Figure 1 in the revised manuscript (see **Figure IV** above, and **Fig. 1a** in the revised manuscript).

Figure 4: Please indicate the position of site I and site II

Following reviewer’s request, we have added the “site I” and “site II” labels to indicate the positions of site I and site II in the updated Figure 4 in the revised manuscript (see **Fig. 4a** in the revised manuscript).

Figure 5: The immunofluorescence localisation of the Myosin VI CBD (1060-1285) and Tom1 shown in figure 5 is not convincing. Both proteins should localise to numerous APPL1/Rab5 positive endosomes in the cell periphery. However, the images in figure 5 show large punctae distributed throughout the cell concentrating in the perinuclear region, possibly cellular protein aggregates. These experiments should be repeated to show the correct localisation of Tom1 and Myosin VI to APPL1 endosomes in the cortical actin network. The biochemical characterisation of the mutant proteins shown in figure 5g is far more convincing and it raises the question whether the experiments shown in figure 5 a-f are required. For the statistical analysis in figure 5f an Anova test and not a student t-test should be used.

We thank the reviewer for this constructive comment, as it has been valuable in improving the scientific quality of the manuscript. To clarify reviewer’s concern, we have repeated the cellular co-localization assay, and indeed, we have found that the large puncta formation is likely due to the cellular protein aggregates. Based on reviewer’s

comment that our biochemical characterization of the mutant proteins shown in Figure 5g is far more convincing and it raises the question whether the experiments shown in figure 5a-f are required, therefore we have decided to remove the original Figure 5a-f, Supplementary Fig. 7, and the related descriptions in the revised manuscript.

Figure 6. Most of the results shown in this figure including the model are not novel. Figure 6a: Coimmunoprecipitation of myosin VI and the autophagy receptors has been published previously. Figure 6b: Simultaneous binding of Tom1 and NDP52 to the CBD of myosin VI was previously shown in Arden et al. *Biochemical Journal*, 2016; Figure 5F. Finally, a very similar model was proposed in Fig. 4C in Tumbarello et al. *JCS* 2013 highlighting that myosin VI can act as a tether that links endosomes via Tom1 and the autophagy adaptors to autophagosomes.

We understand reviewer's concern and thank the reviewer for the above critical comments. We agree with reviewer's comment that the co-immunoprecipitation of Myosin VI and the autophagy receptors has been reported previously. Therefore, we have moved the original Figure 6a to the Supporting Information part in the revised manuscript. However, we don't agree with reviewer's viewpoint that based on the GST-pull down data shown in the Figure 5F of a previous report (Arden et al. *Biochemical Journal*, 2016)⁶ (see **Figure VIIa** below), they can confidently draw the conclusion that Tom1 and NDP52 can simultaneously bind to Myosin VI. As in their GST-pull down assay, excess amounts of GST-Myosin VI CBD are used (see **Figure VIIa** below). Given that both Tom1 and NDP52 can bind to Myosin VI CBD, when in the presence of excess amounts of GST-Myosin VI CBD proteins, definitely both Tom1 and NDP52 can be pulled down with GST-Myosin VI CBD, but they can't rule out the possibility that Tom1 and NDP52 might bind to different populations of Myosin VI CBD forming different complexes. In contrast, in our Co-IP assays, in order to avoid this issue, we have used anti-Flag antibody conjugated beads to precipitate Flag-Tom1 rather than using anti-GFP antibody conjugated beads to precipitate GFP-Myosin VI (see **Figure VIIb** below, and **Fig. 5b** in the revised manuscript). Meanwhile, our control Co-IP data had proved that Tom1 can't directly bind to autophagy receptors, TAX1BP1, NDP52 and Optineurin (see **Supplementary Fig. 8b** in the revised manuscript). However, in the presence of Myosin VI(1060-1285), autophagy receptors TAX1BP1, NDP52, and Optineurin can be readily co-immunoprecipitated with Tom1 (see **Figure VIIb** below, and **Fig. 5b** in the revised manuscript). Therefore, we can surely conclude that Myosin VI can function as a bridging adaptor to mediate the associations of Tom1 with autophagy receptors, TAX1BP1, NDP52 and Optineurin. Furthermore, in our assays, we have tested all the three autophagy receptors not just NDP52. In all, our data are absolutely novel, and are more convincing. As for our proposed model in Figure 6c, it is a structure-based model, a key message from which is to show how Myosin VI associates with Tom1 through the specific interaction between the C-terminal CBD of Myosin VI and the MBM of Tom1. Since the Myosin VI/Tom1 complex structure determined in this study is the first reported atomic structure revealing how Myosin VI interacts with Tom1, therefore it is quite novel. In addition, given that ubiquitin chains may promote the strong associations of autophagy receptors TAX1BP1, NDP52 and Optineurin with Myosin VI in cells, we have also included the ubiquitin chains and updated our model in the revised manuscript (see **Fig. 5d** in the revised manuscript).

Figure VII: The assays used in a previous report and in our study to demonstrate the simultaneous bindings of Tom1 and autophagy receptors to the CBD of Myosin VI. (a) The GST-pull down assay of NDP52 (lane 2) or Tom1 (lane 3) or both (lane 4) with GST-Myosin VI CBD reported in a previous study⁶. **(b)** The co-immunoprecipitation assay in our study revealing that Myosin VI(1060-1285), Tom1 and autophagy receptor TAX1BP1, NDP52 or Optineurin, can form ternary complexes in co-transfected cells.

Minor points

Please correct:

line 83 exocytosis, delete at the Golgi complexes

line 85 that can bind to and walk along actin filaments

line 98 serves

108 was proved to be

109 receptors

110 which are

114 endosomes

115 autophagosomes

121 family members, which can ...

121interact with the C-terminal CBD of Myosin VI

123that the C-terminal CBD of Myosin VI adopts a unique ...

127 receptors

150 ...rather than the N-terminal “RRL” motif region (residues 1032-1157) of the Myosin VI CBD is required ...

384 as their binding sites in the mMyosin VI “RRL” motif are highly overlapped. Not clear, please explain.

Further corrections of the English are required.

We thank the reviewer for his/her careful reading and pointing out our incorrect or confusing writings in the original manuscript, we have fixed all these errors or issues in the revised manuscript. Furthermore, we have taken extra care to improve the writing of our manuscript, and further corrected the English in the revised manuscript.

Specific responses to the criticisms from the Reviewer #3:

We thank the reviewer for the strong supports and constructive suggestions of our manuscript. The following is a list of our detailed responses to the comments raised:

In this manuscript, Hu et al investigated the interaction between the Myosin VI cargo binding domain (CBD) and Tom1, a component of the endosome sorting complex. They discovered, by detailed biochemical and structural characterization, that Myosin VI CBD binds to a short and continuous helix from Tom1. Interestingly, they provided evidences showing that Myosin VI CBD and Tom1 form a 1:1 stoichiometric complex, instead of a 2:2 complex previously characterized for other Myosin VI CBD targets such as Dab2. Formation of the 1:1 stoichiometric complex between Myosin VI CBD and Tom1 suggests that Myosin VI likely functions as a tether between actin filaments and endosomes via binding to Tom1. The authors further showed that the “RRL”-motif of Myosin VI preceding its CBD binds to autophagy receptors like Optineurin, NDP52 and TAX1BP1. With these findings, the authors proposed a model suggesting that Myosin VI tail cargo binding domains can simultaneously tether autophagosome and endosome and promote formation of amphisome. This study provides direct experimental evidence showing that the tail cargo binding domains of Myosin VI may determine the molecular tethering role of the motor via forming a 1:1 heterodimer with its cargo proteins. Therefore, the manuscript is interesting for understanding cellular functions of Myosin VI and also other unconventional myosins. As such, the current manuscript is a strong candidate for NC.

However, there are a number of issues will need to be addressed before the manuscript can be accepted for publication:

1. The evidence that the regions outside aa 437-463 are not involved in binding to Myosin VI CBD is not very strong. The heat releases shown in Fig. 1C&E are significantly different, suggesting that residues N-terminal to 437-463 may also be involved. The author should show both entropy and enthalpy values of their ITC titration curves, as these values will provide evidence whether regions outside 437-463 may also be involved in binding.

We thank the reviewer for this insightful comment. Following reviewer’s suggestion, we have included the entropy values, enthalpy values, the binding stoichiometry N values, and the related Gibbs energy changes derived from our ITC titration results in the revised manuscript (see **Table 1** in the revised manuscript). Meanwhile, we have also repeated the ITC experiment to measure the interaction between Tom1(392-437) and Myosin VI(1157-1285), of which the Tom1(392-437) and Myosin VI(1157-1285) proteins are prepared in an identical buffer condition. The obtained result confirms that there is no ITC-detectable interaction between Tom1(392-437) and Myosin VI(1157-1285) (see **Figure VIIIa** below). In addition, further analytical ultracentrifugation and multi-angle light scattering analyses show that similar to that of Tom1(437-463), Tom1(392-463) forms a monomer and interacts with Myosin VI(1157-1285) to form a 1:1 stoichiometric complex in solution (see **Figure VIIIa** and **VIIIb** below). Interestingly, our NMR experiment using ¹⁵N-labelled Tom1(392-493) titrated with the un-labelled Myosin

VI(1157-1285) reveals that some residues N-terminal to 437-463 of Tom1, such as T412, C415, T426 and D427, display slow exchange chemical shift changing patterns, which are quite different from that of the MBM region (see **Figure Xa** below, and **Supplementary Fig. 7a** in the revised manuscript), suggesting that this N-terminal region may strongly bind to Myosin VI(1157-1285) or undergo a passive conformational rearrangement induced by Tom1 MBM, which is unstructured in its apo-form but forms a continuous α -helix when binding to Myosin VI. Considering that we can't detect the interaction between Tom1(392-437) and Myosin VI(1157-1285) by the ITC analysis and the measured K_d values of Tom1(437-463), Tom1(392-463) and Tom1(392-493) are very similar (see **Table 1** in the revised manuscript), it is more likely that the region N-terminal to Tom1 MBM may undergo a conformational rearrangement induced by the Myosin VI-bound Tom1 MBM, thereby absorbing additional heat to change the heat release profile of Tom1(392-463) and Myosin VI(1157-1285) interaction measured by ITC analysis.

Figure VIII: Biochemical characterizations of the interactions of Myosin VI(1157-1285) with Tom1(392-437) and Tom1(392-463). (a) A ITC-based measurement of the interaction between Tom1(392-437) and Myosin VI(1157-1285). ‘N.D.’ stands for that the K_d value is not detectable. (b) Overlay plot of the sedimentation velocity data of Myosin VI(1157-1285) (red), Tom1(392-463) (green) and the Myosin VI(1157-1285)/Tom1(392-463) complex (black). (c) Overlay plot of the multi-angle light scattering data of Myosin VI(1157-1285), Tom1(392-463), and the Myosin VI(1157-

1285)/Tom1(392-463) complex. The derived molecular masses of Myosin VI(1157-1285) and Tom1(392-463) are shown in red and in green respectively, while the derived molecular mass of the Myosin VI(1157-1285)/Tom1(392-463) complex is shown in black. The molecular masses errors are the fitted errors obtained from the data analysis software, and are showed in the brackets.

2. The NMR data shown in Figure S6 indicated that residues C-terminal to 437-463 are involved in binding to CDB as obvious chemical shift differences are observed. The NMR data indicate that this binding is rather weak but may be important.

We understand reviewer's concern, and indeed, there are some chemical shift differences between the two ^1H - ^{15}N HSQC spectra of Myosin VI(1157-1285) saturated with Tom1(392-463) and Tom1(392-493). However, given that different Tom1 fragment proteins with not identical buffer conditions were used for these NMR titration experiments, these chemical shift differences are not very significant. It is also worthwhile to note that the Myosin VI-binding interface residues N460, L461 and S462 are extremely close to the extreme C-terminal of Tom1(392-463) (see **Fig. 3c** in the revised manuscript), but in contrast, they are just located in the middle region of Tom1(392-493), therefore the local chemical environments generated by these interface residues in Tom1(392-463) and Tom1(392-493) when binding to Myosin VI(1157-1285) might be slightly different. Further, the NMR peak corresponding to the side chain of Myosin VI W1192 residue that is located at the site II of the C-terminal CBD of Myosin VI, displayed little chemical shift changes in the presences of Tom1(392-463) and Tom1(392-493) (see **Supplementary Fig. 6b** in the revised manuscript), but in contrast, it displayed significant chemical shift changes when binding to Dab2¹. To further clarify reviewer's concern, we have constructed two additional Tom1 fragments, Tom1(437-493) and Tom1(464-493), and quantitatively measured their interactions with Myosin VI(1157-1285) by ITC analyses. The ITC results show that the Tom1(437-493) fragment binds to Myosin VI(1157-1285) with a binding affinity *Kd* value, about 0.76 μM (see **Figure IXa** below, and **Supplementary Fig. 3d** in the revised manuscript), which is very similar to that of the Tom1(437-463) fragment (see **Figure IXb** below, and **Fig. 1e** in the revised manuscript), while Tom1(464-493) can't interact with Myosin VI(1157-1285) (see **Figure IXc** below, and **Supplementary Fig. 3e** in the revised manuscript). In addition, our NMR titration experiment using ^{15}N -labelled Tom1(392-493) titrated with the un-labelled Myosin VI(1157-1285) reveals that the MBM region of Tom1 (residues 440-462) shows significant dose-dependent peak-broadenings, while the region C-terminal to MBM (residues 463-493) displays negligible chemical shift changes (see **Figure X** below, and **Supplementary Fig. 7** in the revised manuscript). Taken all data together, we conclude that the residues C-terminal to the MBM of Tom1 are not involved in the binding to Myosin VI(1157-1285). We have included these data in the revised manuscript.

Figure IX: The region C-terminal to the MBM of Tom1 are not involved in the binding to Myosin VI(1157-1285). (a-c) ITC-based measurements of the binding affinities of Myosin VI(1157-1285) with Tom1(437-493) (a), Tom1(437-463) (b), and Tom1(464-493) (c). K_d values are the fitted dissociation constants with standard errors, when using the one-site binding model to fit the ITC data. ‘N.D.’ stands for that the K_d value is not detectable.

3. It will probably be more sensitive to titrate unlabelled Myosin VI CBD to ^{15}N -labelled Tom1 392-493 (or other fragments if more suitable) to investigate whether residues outside aa 437-463 may be involved in Myosin VI binding by careful NMR-based investigations.

This is a nice suggestion. Following reviewer’s comments, we have purified the ^{15}N -labelled Tom1(392-493) and titrated with unlabelled Myosin VI(1157-1285) proteins. Consistent with the data from the ^{15}N -labelled Tom1(392-463), titration of the ^{15}N -labelled Tom1(392-493) fragment with Myosin VI(1157-1285) shows that a select set of peaks in the ^1H - ^{15}N HSQC spectrum of Tom1(392-493) undergo significant dose-dependent peak-broadenings (see **Figure Xa** below, and **Supplementary Fig. 7a** in the revised manuscript), confirming that this Tom1 fragment can specifically bind to Myosin VI(1157-1285). After backbone chemical shift assignments, we have plotted the amide backbone chemical shift changes and peak-broadenings as a function of the residue number of Tom1(392-493), and found that the significant backbone amide peak-broadenings are only rich in the MBM region of Tom1 (residues 440-462), while other regions either N-terminal or C-terminal to MBM display small chemical shift changes (less than 0.018 ppm) (see **Figure Xb** below, and **Supplementary Fig. 7b** in the revised manuscript). Interestingly, some residues located in a region N-terminal to MBM of Tom1, such as T412, C415, T426 and D427, display slow exchange chemical shift changing patterns (see **Figure Xa** below, and **Supplementary Fig. 7a** in the revised manuscript), indicating that this N-terminal region may strongly bind to Myosin VI(1157-1285) or undergo a passive conformational rearrangement induced by Tom1 MBM, which is unstructured in its apo-form but forms a continuous α -helix when binding to Myosin VI. Given that we can’t detect the interaction between Tom1(392-437) and Myosin VI(1157-1285) by ITC (see **Figure VIIIa** above), and the ITC-measured K_d values of Tom1(437-463), Tom1(392-463), and Tom1(392-493) are very similar (see

Table 1 in the revised manuscript), it is more likely that the region N-terminal to Tom1 MBM may undergo a conformational rearrangement induced by the Myosin VI-bound Tom1 MBM rather than directly binding to Myosin VI(1157-1285). Nevertheless, we will clarify this issue in a following-up manuscript.

Figure X: NMR-based characterizations of the interaction between Tom1(392-493) and Myosin VI(1157-1285). (a) Superposition plots of the ^1H - ^{15}N HSQC spectra of ^{15}N -

labelled Tom1(392-493) titrated with the increasing molar ratios of un-labelled Myosin VI(1157-1285) proteins. **(b)** A plot of backbone amide chemical shift differences and peak-broadenings as a function of the residue number of Tom1(392-493) between the wild-type and the protein titrated with Myosin VI(1157-1285) at the molar ratio of 2:1. In this representation, the residues with disappeared NMR peaks due to peak-broadenings are shown in black, and the combined ^1H and ^{15}N chemical shift changes are defined as:

$$\Delta_{\text{ppm}} = [(\Delta\delta_{\text{HN}})^2 + (\Delta\delta_{\text{N}} \times \alpha_{\text{N}})^2]^{1/2}$$

Where $\Delta\delta_{\text{HN}}$ and $\Delta\delta_{\text{N}}$ represent chemical shift differences of amide proton and nitrogen chemical shifts of the each residue of Tom1(392-493). The scaling factor (α_{N}) used to normalize the ^1H and ^{15}N chemical shift is 0.17.

4. The authors have identified single point mutations in Tom1 which can abolish the Myosin VI/Tom1 interaction. It will add a lot value to the current manuscript if the authors can demonstrate that such Tom1 mutants are defective in amphisome formation as proposed by the authors.

We understand reviewer's concern, and thank the reviewer for the above constructive suggestions. Actually, we also seek to perform some functional assays based on our structural data to evaluate the roles of different Tom1 variants in amphisome formation. But due to our technical limitation, we can't get a suitable Tom1 knock-out cell-line to carry out this type of functional assay. Meanwhile, unfortunately, we can't find a good collaborator to test this type of functional assay with respect to amphisome formation in a limited time for revision. Thus, we are unable to include any functional data in this manuscript. Given that the essential roles of Tom1 and Myosin VI for promoting the maturation of autophagosome are well documented in previous functional studies conducted by other groups^{7,8}, therefore in this manuscript we just focus on the biochemical and structural results, which, in our opinions, are the major strength of our manuscript.

5. The authors would need to explain how the extremely weak bindings between the Myosin VI "RRL"-motif with autophagy receptors may fulfill the motor's tethering role with autophagosome.

See our detailed response to the point #5 from the Reviewer 2.

References

1. Yu, C. et al. Myosin VI undergoes cargo-mediated dimerization. *Cell* **138**, 537-48 (2009).
2. He, F.H. et al. Myosin VI Contains a Compact Structural Motif that Binds to Ubiquitin Chains. *Cell Reports* **14**, 2683-2694 (2016).
3. Li, F.X. et al. Structural insights into the ubiquitin recognition by OPTN (optineurin) and its regulation by TBK1-mediated phosphorylation. *Autophagy* **14**, 66-79 (2018).
4. Hu, S.C. et al. Mechanistic Insights into Recognitions of Ubiquitin and Myosin VI by Autophagy Receptor TAX1BP1. *Journal of Molecular Biology* **430**, 3283-3296 (2018).

5. Xie, X.Q. et al. Molecular basis of ubiquitin recognition by the autophagy receptor CALCOCO2. *Autophagy* **11**, 1775-1789 (2015).
6. Arden, S.D., Tumbarello, D.A., Butt, T., Kendrick-Jones, J. & Buss, F. Loss of cargo binding in the human myosin VI deafness mutant (R1166X) leads to increased actin filament binding. *Biochemical Journal* **473**, 3307-3319 (2016).
7. Tumbarello, D.A. et al. Autophagy receptors link myosin VI to autophagosomes to mediate Tom1-dependent autophagosome maturation and fusion with the lysosome. *Nat Cell Biol* **14**, 1024-35 (2012).
8. Verlhac, P. et al. Autophagy Receptor NDP52 Regulates Pathogen-Containing Autophagosome Maturation. *Cell Host & Microbe* **17**, 515-525 (2015).

Reviewers' Comments:

Reviewer #1:

Remarks to the Author:

All of my concerns have been addressed and the manuscript is much improved. The science is certainly of high quality and ready for publication. It still needs a read-through by a native English speaker but the content is easy to follow.

Reviewer #2:

Remarks to the Author:

The authors have made substantial efforts to address my concerns and have improved the the mnauscript accordingly. I therefore recommend publication of the manuscript in its current much improved form.

Reviewer #3:

Remarks to the Author:

The authors have done an extensive list of experiments to address the comments raised by the reviewers. This reviewer is satisfied with the responses from the authors to my questions and requests. I understand that the authors are not in the position to perform functional experiments raised in my review (point #4). My original intention was to push the authors for such functional experiment with anticipation of potential difficulty. The revised manuscript is a quite complete structural and mechanistic study of the interaction between myosin VI and Tom1, and I support publication of the work in NC.

Point-by-point responses to the comments raised by the referees
(Referees' comments are in **blue**, and our responses are in **black**)

Specific responses to the comments from the Reviewer #1:

All of my concerns have been addressed and the manuscript is much improved. The science is certainly of high quality and ready for publication. It still needs a read-through by a native English speaker but the content is easy to follow.

We thank the reviewer for his/her wonderful supports.

Specific responses to the comments from the Reviewer #2:

The authors have made substantial efforts to address my concerns and have improved the manuscript accordingly. I therefore recommend publication of the manuscript in its current much improved form.

We thank the reviewer for his/her strong supports of our manuscript.

Specific responses to the comments from the Reviewer #3:

The authors have done an extensive list of experiments to address the comments raised by the reviewers. This reviewer is satisfied with the responses from the authors to my questions and requests. I understand that the authors are not in the position to perform functional experiments raised in my review (point #4). My original intention was to push the authors for such functional experiment with anticipation of potential difficulty. The revised manuscript is a quite complete structural and mechanistic study of the interaction between myosin VI and Tom1, and I support publication of the work in NC.

We appreciate reviewer's constructive suggestions, and thank reviewer's strong supports of our manuscript.